

# The Mixed Layer Depth in the Ocean Model Intercomparison Project (OMIP): Impact of Resolving Mesoscale Eddies

Anne Marie Treguier[1], Clement de Boyer Montégut[1], Alexandra Bozec[2], Eric P. Chassignet[2], Baylor Fox-Kemper[3], Andy McC. Hogg[4], Doroteaciro Iovino[5], Andrew E. Kiss[4], Julien Le Sommer[6], Yiwen Li[7], Pengfei Lin[7], Camille Lique[1], Hailong Liu[7], Guillaume Serazin[1], Dmitry Sidorenko[8], Qiang Wang[8], Xiaobio Xu[2], Steve Yeager[9]

[1]Univ Brest, CNRS, Ifremer, IRD, Laboratoire d'Océanographie Physique et Spatiale (LOPS), IUEM, Brest 29280, France
[2]Center for Ocean–Atmospheric Prediction Studies, Florida State University, Tallahassee, USA
[3]Department of Earth, Environmental, and Planetary Sciences, Brown University, Providence, RI, USA
[4]Research School of Earth Sciences and ARC Centre of Excellence for Climate Extremes, Australian National University, Canberra, Australia
[5]Ocean Modeling and Data Assimilation Division, Fondazione Centro Euro-Mediterraneo sui Cambiamenti Climatici (CMCC), Bologna, Italy
[6]Univ. Grenoble Alpes, CNRS, IRD, Grenoble INP, INRAE, IGE, Grenoble, France
[7]State Key Laboratory of Numerical Modeling for Atmospheric Sciences and Geophysical Fluid Dynamics, Institute of Atmospheric Physics, Chinese Academy of Sciences, Beijing, China
[8]Alfred Wegener Institute, Helmholtz Centre for Polar and Marine Research (AWI), Bremerhaven, Germany
[9]National Center for Atmospheric Research, Boulder, CO, USA

*Correspondence to*: Anne Marie Treguier (anne-marie.treguier@univ-brest.fr)

**Abstract**

The ocean mixed layer is the interface between the ocean interior and the atmosphere or sea ice, and plays a key role in climate variability. It is thus critical that numerical models used in climate studies are capable of a good representation of the mixed layer, especially its depth. Here we evaluate the mixed layer depth (MLD) in six pairs of non-eddying (1° resolution) and eddy-rich (up to 1/16°) models from the Ocean Model Intercomparison Project (OMIP), forced by a common atmospheric state. For model validation, we use an updated MLD dataset computed from observations using the OMIP protocol (a constant density threshold). In winter, low resolution models exhibit large biases in the deep water formation regions. These biases are reduced in eddy-rich models but not uniformly across models and regions. The improvement is most noticeable in the mode water formation regions of the northern hemisphere. Results in the Southern Ocean are more contrasted, with biases of either sign remaining at high resolution. In eddy-rich models, mesoscale eddies control the spatial variability of MLD in winter. Contrary to a hypothesis that the deepening of the mixed layer in anticyclones would make the MLD larger globally, eddy-rich models tend to have a shallower mixed layer at most latitudes than coarser models do. In addition, our study highlights the sensitivity of the MLD computation to the choice of a reference level and the spatio-temporal sampling, which motivates new recommendations for MLD computation in future model intercomparison projects.





## 1 Introduction

The ocean mixed layer is the interface between the ocean interior and the atmosphere or sea ice. It is a layer of thickness ranging from a few meters to hundreds of meters, homogenized in the vertical by wind, buoyancy and wave- driven turbulence (D'Asaro, 2014). Because of the existence of this turbulent layer, fluxes from the atmosphere or sea ice modify the ocean properties not only at the surface, but also over the thickness of the mixed layer. This makes the mixed layer depth (MLD) a key variable for Earth's climate, as it controls the relationship between air-sea fluxes and sea surface temperature, and thus influences climate feedback mechanisms. In the mixed layer, potential density is relatively homogeneous, compared to the stratification below.

The density stratification (pycnocline) that begins at the base of the mixed layer can be due to vertical gradients of temperature or salinity (Helber et al., 2012). Most often, the mixed layer base is formed where the cooler pycnocline meets the warmer waters directly heated by air-sea fluxes, and where deep ocean heat uptake on longer than seasonal timescales is affected by mixed layer detrainment. In the tropics and polar seas, "barrier layers" occur when the mixed layer is fresh and the pycnocline just below is due to the salinity gradient, while the temperature profile remains relatively homogeneous vertically (in the tropics, de Boyer Montégut et al., 2007; MacKinnon et al., 2016; Mignot et al., 2007), or when temperature does not impact density much in comparison to salinity (in the polar regions: MacKinnon et al., 2016; Pellichero et al., 2017; Peralta-Ferriz & Woodgate, 2015). Barrier layers are shown to influence the development of tropical cyclones (Rudzin et al., 2018). Most of the ocean primary production takes place in the surface mixed layer, which often coincides with the euphotic zone, and thus the MLD is also important for ecosystem functioning and the ocean uptake of carbon (Llort et al., 2019; Uchida et al., 2020). The MLD is highly variable in space and time (de Boyer Montégut et al., 2004; Holte et al., 2017), and presents a strong seasonal cycle. Fig. 1 shows a climatology of the mixed layer in winter and summer from observations (de Boyer Montégut, 2022). The winter deepening of the mixed layer at mid to high latitudes is highly heterogeneous in space, with well-known regions of large MLD related to deep water formation (Labrador Sea and Greenland Sea for example, Schulze et al., 2016). At smaller spatial scales, mesoscale eddies and fronts have an impact on the MLD, as shown recently from observations (Gaube et al., 2019; Hausmann et al., 2017; Shroyer et al., 2020). The mixed layer is also highly variable on diurnal and synoptic timescales, as night time cooling or wind and wave events can drive significant deepening, while the mixed layer may quickly restratify during calm, warm periods through solar or dynamical mechanisms (Haney et al., 2012; Q. Li et al., 2017; Q. Li & Fox-Kemper, 2020). This lends a profoundly irreversible aspect to mixed layer variability and connects high-frequency forcing to seasonal to decadal evolution of the layer by rectification.



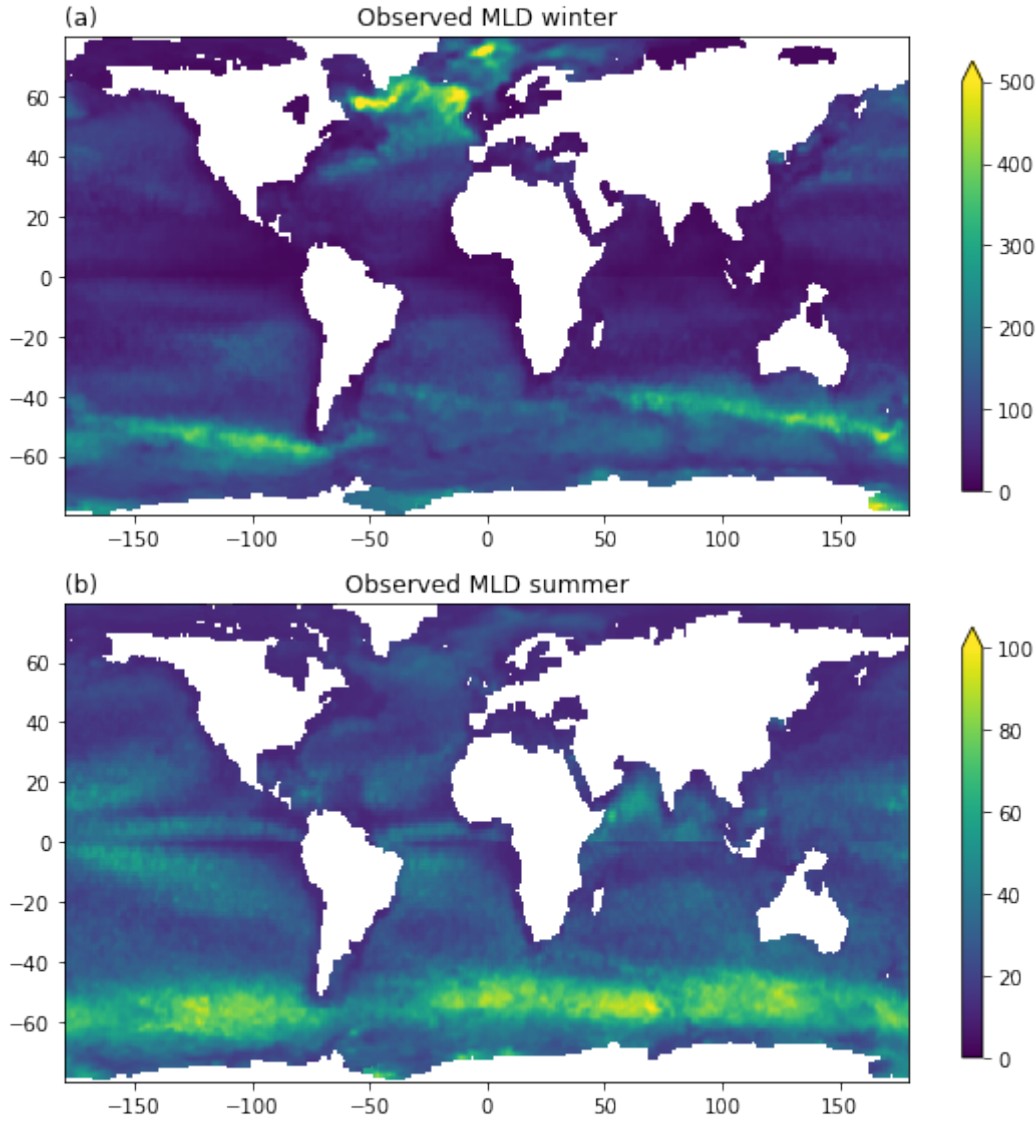

**Figure 1 : Observed MLD (m) in winter (a) and summer (b) (de Boyer Montégut, 2022). The winter (summer) MLD is the average over the months of January to March in the Northern (Southern) Hemisphere, and July to September in the Southern (Northern) Hemisphere, respectively. Note the differing color scales for the two seasons.**

For reliable projections of the future climate, the mixed layer depth and its variability need to be well represented in numerical models (Semmler et al., 2021), but it is not the case presently (Belcher et al., 2012; Fox-Kemper et al., 2021; Q. Li et al., 2019; Pan et al., 2023; Sallée et al., 2013). Analysis of the coupled Model Intercomparison projects (CMIP) have revealed systematic biases: mixed layers were found too shallow in summer in CMIP5 (C. J. Huang et al., 2014; Sallée et al., 2013) and too deep in winter in CMIP5 and CMIP6 in some regions of deep water formation (Fox-Kemper et al., 2021; Heuzé, 2021). In future



projections using CMIP5 models, Alexander et al. (2018) find that the summer MLD decreases with strong anthropogenic forcing, a tendency that does not fit the trends observed during the historical period period (Sallée et al., 2021) but is evident

in the CMIP6 ensemble as well (Fox-Kemper et al., 2021).

A vast effort is currently underway to increase the spatial resolution of ocean climate models, in order to resolve mesoscale fronts and eddies, notably in the context of HighResMIP (Haarsma et al., 2016). Positive impacts of resolving ocean eddies on the dynamics of western boundary currents, equatorial currents, and the Antarctic Circumpolar Current have been demonstrated in forced ocean models (Chassignet et al., 2020; Hecht and Hasumi, 2008) as well as in the recent HighResMIP

coupled models (Beech et al., 2022; Grist et al., 2018; Roberts et al., 2018, 2020) for the representation of these currents and heat transports. However, Chassignet et al. (2020) did not find systematic improvements in salinity and temperature metrics with resolution. No assessment has been made yet regarding the representation of the MLD at the global scale. Evaluating mixed layer characteristics in fully coupled models is a difficult task, because the MLD depends not only on ocean characteristics (ocean circulation, parameterization of waves, turbulence and vertical convection in each model), but also on

the atmosphere and its variability (winds, air temperature, clouds…); these differ considerably across CMIP models.

The Ocean Model Intercomparison Project (OMIP, Griffies et al., 2016) provides an ideal framework to evaluate the impact of ocean model resolution on the MLD, which is the purpose of our study. OMIP makes use of two common atmospheric and river runoff datasets to drive global ocean-sea ice models. OMIP phase 1 (Griffies et al., 2009) was forced by the COREII dataset (Large and Yeager, 2009), which is mainly derived from the NCEP atmospheric reanalysis and covers a period of 62

years (1948–2009). The second phase (OMIP2) is based on the more recent JRA55-do forcing derived from the Japanese 55 years Reanalysis (Griffies et al., 2016; Tsujino et al., 2018) which covers the period 1958-2018. The increased temporal frequency and refined horizontal resolution makes JRA55-do more appropriate to force eddy resolving ocean models. Chassignet et al. (2020) have used four pairs of OMIP2 simulations, integrated for one forcing cycle, to evaluate the impact of horizontal resolution on ocean kinetic energy, temperature, salinity, sea level, sea ice, meridional overturning circulation and

Drake passage transport. The model pairs included a low resolution (typically 1°) and a high resolution member (typically 1/10°) with mostly comparable parameterization settings in each pair. A wide range of model variables have been assessed and compared with observations (temperature, salinity, sea surface height, eddy kinetic energy, sea ice), showing improvements at high resolution in some (but not all) of them.

We use the same experimental protocol as Chassignet et al. (2020), but additional models are included (up to 1/16° resolution,

based on Iovino et al., 2016). The intercomparison of low resolution models has shown that the MLD is very model-dependent (Tsujino et al., 2020). This provokes several questions, which we address here: is the MLD model-dependent in eddy-rich models? Are there improvements of the simulated MLD in regions of high mesoscale variability? The MLD is observed to be deeper in anticyclones (Gaube et al., 2019), although Hausmann et al. (2017) suggest that the net effect at large scales may be small. Is the MLD systematically deeper in eddy-rich models, compared to non-eddying models?

The MLD is a nonlinear function of the density profile, and its statistics are not gaussian (Johnson & Lyman, 2022), both of which create methodological difficulties for the evaluation of the models against observations as well as for model



intercomparisons. This motivates an investigation of the influence of spatio-temporal sampling and MLD computation algorithms in section 3, following the presentation of the models in section 2. Section 4 presents the influence of resolution on the MLD biases at the global scale, and section 5 focuses on water mass formation regions. Conclusions are presented in
section 6, where we also discuss an update to the OMIP/CMIP protocol regarding the diagnostic of the MLD that was proposed by Griffies et al. (2016).

## 2 Description of the model pairs

The spin up of the deep ocean occurs on centennial time scales (Griffies et al., 2009), which is why the OMIP protocol requires repeating the JRA55-do (years 1958 to 2018) forcing for 6 consecutive cycles (Tsujino et al., 2020). However such long
simulations are usually too costly in computing time for the high resolution models. Here, as in Chassignet et al. (2020), we consider only the first OMIP2 cycle, which is adequate for processes near the ocean surface. Only the last 30 years are analyzed to reduce spin-up effects.

Table 1 summarizes some features of the model pairs, relevant for our study. We use six model pairs, including the four model pairs described in Chassignet et al. (2020), whose naming convention (institution-ocean model name) we follow. Note that
this naming convention differs from CMIP where a single "source name" is used for each model, because some datasets used here are not published on the Earth System Grid Federation (ESGF). When relevant, the ESGF source name is also indicated in Table 1. FSU-HYCOM is a global configuration of the HYbrid Coordinate Ocean Model (HYCOM, Chassignet et al., 2003). Vertical mixing is parameterized by the K-profile parameterization (KPP, Large et al., 1994). The NCAR-POP model is based on the Parallel Ocean Program POP (Smith et al., 2010). It is the global ocean component of the Community Earth System
Model version 2 (CESM2, Danabasoglu et al., 2020) and the high resolution version is documented by Chang et al. (2020). Vertical mixing is parameterized by KPP. There are two other parameterizations, targeted at mixed layer dynamics, used in the low resolution version of POP but not the high-resolution version: a parameterisation of submesoscale eddy effects (Fox-Kemper et al., 2008, hereafter FFH; Fox-Kemper et al., 2011) and a parameterization of Langmuir turbulence (Q. Li et al., 2016). AWI-FESOM is the Finite element/volumE Sea ice-Ocean Model (FESOM) version 1.4 (Wang et al., 2014). It differs
from the other models considered here because of its unstructured grid. The low resolution version has 1° in most regions, up to 25 km in the polar seas and 30 km at the equator (0.127 million grid nodes on the horizontal); the high resolution version has a grid scaled by the observed sea surface height variance, from 10 km in areas of high eddy activity to about 50 km elsewhere (1.3 million surface grid nodes; Sein et al., 2017). Maps of the grid resolution are shown in Fig. 1 of Chassignet et al. (2020). Vertical mixing is also represented by the KPP parameterization in this model. IAP-LICOM is a global configuration
of the LASG/IAP Climate system Ocean Model (LICOM) version 3, developed in the Laboratory of Atmospheric Sciences and Geophysical fluid dynamics (LASG) of the Institute of Atmospheric Physics (IAP), Chinese Academy of Sciences (L. Li et al., 2020; Lin et al., 2020). Vertical mixing in the mixed layer for both momentum and tracers is computed by the scheme



of Canuto et al., (2001, 2002) with an upper limit of $2 \cdot 10^{-2}$ $m^2.s^{-1}$. We do not include more details about these four model pairs, because they are documented extensively by Chassignet et al. (2020).

ACCESS-MOM is the ocean component of the Australian Community Climate and Earth System Simulator (ACCESS) developed by a consortium of Australian universities and research institutes. It consists of a MOM5.1 ocean (Griffies, 2012) coupled to the CICE5.1.2 sea-ice model (Hunke et al., 2015) at 1° and 0.1° nominal horizontal resolution. These are updates (see supporting information in Solodoch et al., 2022) of the configurations described by Kiss et al. (2020) and the 1° configuration is the ocean component of the ACCESS-CM2 climate model (Bi et al., 2020). The 0.1° grid is Mercator within

65° of the equator, tripolar north of 65°N and has uniform meridional spacing south of 65°S. The 1° grid is similar but with a refinement to 1/3° meridional spacing within 10° of the equator. The vertical coordinate is z*, with 75 levels (1.1 m spacing at the surface; 198 m at depth) for the 0.1° resolution and 50 levels (2.3 m spacing at the surface; 220 m at depth) for the 1° resolution, with spacing increasing smoothly with depth to optimize resolution of baroclinic modes (Stewart et al., 2017). Both resolutions use FFH and parameterise vertical mixing by KPP, Simmons et al. (2004) bottom-enhanced internal tidal mixing

and Lee et al. (2006) barotropic tidal mixing. Both resolutions include background vertical diffusivity, a constant $10^{-6}$ $m^2.s^{-1}$ at 0.1° resolution but increasing from $10^{-6}\,m^2.s^{-1}$ at the equator to a constant $5. \cdot 10^{-6}\,m^2.s^{-1}$ poleward of 20° (Jochum, 2009) at 1° resolution. The 1° configuration also has down-slope mixing (Beckmann & Döscher, 1997; Campin & Goosse, 1999; Döscher & Beckmann, 2000).

CMCC-NEMO at low resolution is the ocean component of the CMCC climate model (CMCC-CM2, Cherchi et al., 2019),

that is based on the Community Earth System Model (CESMv1.2), in which the original ocean component is replaced by the Nucleus for European Modelling of the Ocean (NEMO) version 3.6 (Madec & the NEMO team, 2016) that is coupled to the Community Ice Code CICEv4.1 (Hunke & Lipscomb, 2008) via the cpl7 coupling architecture. Discretization is performed on a tripolar quasi-isotropic grid with nominal resolution 1°, refined poleward following a Mercator projection and refined in latitude near the equator (1/3°). The vertical grid has 50 levels, with a thickness of 1m at the surface increasing to 400m at

depth. An iso-neutral lateral diffusivity, with a coefficient varying as the grid spacing, is used together with an eddy-induced velocity with variable coefficient. Vertical mixing of momentum and tracers is performed by the TKE (Turbulent Kinetic Energy) parameterization introduced by Blanke & Delecluse (1993), modified since then in NEMO to include Langmuir Cells and the influence of surface wave breaking (Madec & the NEMO team, 2016). The high resolution configuration version is largely based on its first implementation described in Iovino et al. (2016). It is based on NEMO version 3.64 coupled with the

Louvain la Neuve Ice Model (LIM) version 2 (Bouillon et al., 2009). It makes use of a nonuniform tripolar grid with a 1/16° horizontal resolution, which is 6.9 km at the equator, and increases poleward as cosine of latitude (minimum grid spacing is ~2 km around Victoria Island in the Arctic Ocean and a constant 3 km south of 60°S). There are 98 vertical levels, with a grid spacing of 1 m near the surface and 160 m at depth. Lateral mixing is parameterized by biharmonic viscosity and diffusion with a coefficient varying as the cube of the grid size. Vertical mixing uses the same TKE parameterization as in the low

resolution configuration.



| Model (ESGF source name) | Horizontal grid | Vertical grid | Parameterizations in the mixed layer | Online MLD : method and reference level |
|---|---|---|---|---|
| ACCESS-MOM (ACCESS-OM2). Low resolution | 1° tripolar | 50 levels<br>Top level: 1.15 m | KPP<br>FFH | Buoyancy threshold 0.0003 ms$^{-2}$<br>Ref: top model level |
| ACCESS-MOM High resolution | 1/10° tripolar | 75 levels<br>Top level: 0.55 m | KPP<br>FFH | Same as low res |
| AWI-FESOM (AWI-CM-1-1-LR) Low resolution | Unstructured grid, nominal 1°, up to 25 km | 46 levels<br>Top level: 0 m | KPP | Density threshold 0.125 kg.m$^{-3}$.<br>Ref: top model level |
| AWI-FESOM (AWI-CM-1-1-MR) High Resolution | Unstructured grid, 10 to 50 km | 46 levels<br>Top level: 0 m | KPP | Density threshold 0.125 kg.m$^{-3}$.<br>Ref: top model level |
| IAP-LICOM (FGOALS-f3-L) Low resolution | 1° tripolar | 30 levels<br>Top level: 5 m | Canuto MLD scheme. | Temperature threshold 0.1°<br>Ref: top model level |
| IAP-LICOM (FGOALS-f3-H) High resolution | 1/10° tripolar | 55 levels<br>Top level: 2.5 m | Canuto MLD scheme | Same as low res |
| NCAR-POP (CESM2) Low resolution | 1° tripolar | 60 levels<br>Top level: 5 m | KPP<br>FFH<br>Langmuir | Density threshold 0.03 kg.m$^{-3}$.<br>Ref: top model level |
| NCAR-POP High resolution | 1/10° tripolar | 62 levels<br>Top level: 5 m | KPP | Max buoyancy gradient (Large et al 1997) |
| FSU-HYCOM Low resolution | 0.72° tripolar | 41 hybrid layers | KPP | Variable density threshold (equivalent 0.3°C)<br>Ref: 1m |



| FSU-HYCOM High resolution | 1/12° tripolar | 36 hybrid layers | KPP | Same as low res |
|---|---|---|---|---|
| CMCC-NEMO (CMCC-CM2-SR5) Low resolution | 1° tripolar | 50 levels Top level: 0.5 m | TKE | Density threshold 0.03 kg.m$^{-3}$. Ref: top model level |
| CMCC-NEMO High resolution | 1/16° tripolar | 98 levels Top level: 0.5 m | TKE | Same as low res |

**Table 1: Model characteristics (see text for more details). Consortia or institution names are as follows: Australian Community Climate and Earth System Simulator (ACCESS), Alfred Wegener Institute (AWI), Florida State University (FSU), Institute of**
**Atmospheric Physics (IAP), National Center for Atmospheric Research (NCAR), Centro Euro-Mediterraneo sui Cambiamenti Climatici (CMCC). Note that the MLD based on the online methods indicated in the last column is not used in this study. This column is intended to show the variety of methods used in model online MLD calculations, which motivated us to recalculate MLD offline (see text).**

As noted by Chassignet et al. (2020), the high resolution models differ from their low resolution counterparts by more than the
horizontal resolution (and the parameterizations of lateral processes that depend on it). For CMCC-NEMO, the sea ice models are not the same and they employ different bulk salinity, affecting the salt release from the sea ice into the ocean. For ACCESS-MOM, IAP-LICOM and CMCC-NEMO, the vertical grid is finer in the high resolution model. However, the parameterizations of vertical mixing are the same at low and high resolution, apart from the down-slope mixing and latitudinal variation of background vertical diffusivity in ACCESS-MOM at low resolution. The MLD has been computed online in the models every
time step and monthly means have been saved. Unfortunately, as shown in Table 1, the computation has not been done following the OMIP protocol of Griffies et al. (2016); rather, each modeling group used a different method, and, in the case of NCAR, the computation is different in the low resolution and high resolution model. These different computation methods have of course an impact on the MLD, which can be avoided by  recomputing the MLD from the monthly averages of temperature and salinity for the intercomparison of the models, but this raises non trivial issues of sampling. These
methodological questions are addressed in the following section.

## 3 Computing the mixed layer depth from observations and models

### 3.1 Defining the depth of the mixed layer

Potential density is expected to be quasi-homogeneous in the mixed layer, and increase downward in the stratified layers
below. Beyond this rather vague statement, the concept of mixed layer is arbitrary (de Boyer Montégut et al., 2004, hereafter



BM04) and as a result, many definitions of the MLD have been proposed in the literature. The MLD may be computed using a threshold change in density or temperature (BM04), a threshold in density gradient (Dong et al., 2008), a maximum density gradient (Large et al., 1997), a maximum in the curvature of the density profile (Lorbacher et al., 2006), a minimum of the relative variance (Huang et al., 2018) or based on energetic principles (Reichl et al., 2022). Before the ARGO network started

in the early 2000s, temperature thresholds were used extensively instead of density thresholds because temperature profile measurements were more numerous than salinity measurements. A threshold on density is more physically relevant because it is the density gradient that hinders the development of turbulence. A density threshold captures both temperature and salinity driven stratifications at the mixed layer base, and includes barrier layers.

However, in some regions there are density-compensated (vertical) gradients of temperature and salinity at the base of the

mixed layer, where a density threshold may overestimate the MLD (BM04). Despite these "vertically compensated layers", the potential density threshold method has been recommended by Griffies et al. (2016) to compute the MLD in OMIP and CMIP models, with a threshold value of 0.03 kg.m$^{-3}$. These authors advocate a uniform constant threshold, even though BM04 have pointed out that the density threshold should vary according to the sea surface temperature (SST). This is because the thermal expansion coefficient of water is smaller when the water is cold. For example, at a temperature of 0°C, a density

change of 0.03 kg.m$^{-3}$ requires a temperature change of 0.6°C, which is a temperature variation greater than one would generally accept in a "well mixed" layer. A spatially variable threshold may be appealing for observations, but less so in the case of numerical models: a threshold dependent on each model's SST would make model intercomparisons more difficult. Also, the complexity of ensuring a spatially-variable threshold was consistently applied in all models and observations is daunting.These considerations led to the choice of a fixed density threshold by Griffies et al. (2016).

MLD definitions other than the threshold method, such as gradient, curvature, or combinations of criteria (Holte & Talley, 2009) have not been proposed for OMIP because of their complexity or because of their possibly strong dependency on the vertical resolution in models with coarse vertical grids. Note however that the criterion of Large et al. (1997) has been used in some high resolution models (Whitt et al., 2019) and that a simplified version of the Holte and Talley algorithm has recently been adapted to numerical models (Courtois et al., 2017).

Near the ocean surface, the density profile varies nonlinearly with depth, and the MLD is also a very nonlinear function of the density profile; these are two important reasons why the computation of the MLD can be strongly method-dependent. Fig. 2 illustrates the sensitivity of MLD to the choice of the density threshold, and how this sensitivity varies seasonally. When a stratified density profile is mixed by winds and waves, without any buoyancy input, the new density of the mixed layer is the depth average over that layer, and a sharp gradient is created below (Fig. 2a). This is the mechanism that generates the so-

called "transition layer". In observations the thickness of this layer is controlled by mixing mechanisms such as shear instabilities and internal wave breaking (Johnston & Rudnick, 2009), and has been estimated to be 23 m on average at the global scale by Serazin et al. (2023). In models, the thickness of this transition layer needs interpretation including aspects of horizontal averaging of eddy features over grid cells as well (Danabasoglu et al., 2008). Because of this sharp density gradient just below the mixed layer in mid-latitude summertime profiles, the MLD is not strongly dependent on the choice of a density



threshold. Fig. 2b shows that the MLDs computed with threshold densities of 0.01 and 0.03 kg.m$^{-3}$ are very similar (blue and

red dots). Density profiles are different in winter: buoyancy is removed from the surface by atmospheric cooling, and deep

mixed layers overlay a weaker stratification (note the different density scale in Figs. 2c and 2d, compared to panels a and b).

In Fig. 2c, the two density thresholds give distinct MLDs. In spring, warming at the surface creates thin stratified layers. Fig.

2d illustrates a case where the re-stratified layer is captured by one density criterion but not the other, leading to very different

MLDs (a difference of order 200 m).

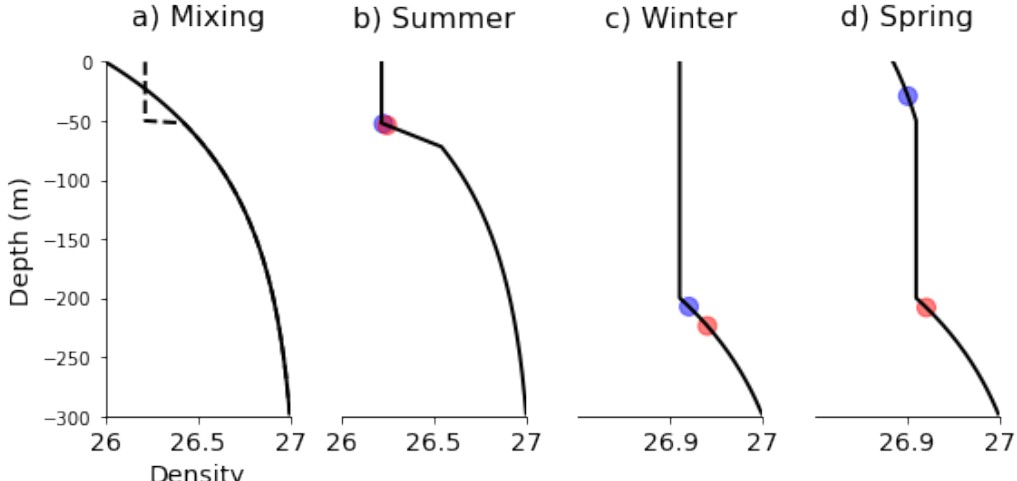

**Figure 2 : Density profiles to illustrate the dependency of the MLD on the density threshold. Blue (red) dots are MLDs computed using a threshold of 0.01 (0.03) kg.m$^{-3}$ respectively. a: an exponential stratified profile (continuous line) is mixed by wind- and wave-generated turbulence down to 50 m depth (dashed lines), in the absence of surface buoyancy input, generating a sharp density**
**gradient. b: typical summer profile resulting from wind and wave mixing. c: typical winter profile resulting from buoyancy loss through surface fluxes, down to 200 m depth. d: spring profile with near-surface restratification.**

### 3.2 Influence of the reference depth

Although the influence of the choice of density or temperature threshold has been extensively examined (e.g. BM04), another

methodological choice has been overlooked, namely, the choice of the depth relative to which the threshold is computed. In

most observation-based studies, a reference depth of 10 m is chosen, to avoid capturing short events of shallow mixed layers

that may occur during the day, but will be mixed down again at night due to surface heat loss and convection (Brainerd &

Gregg, 1995). Quoting BM04: "The MLD we want to estimate is the depth through which surface fluxes have been recently

mixed and so integrated, recently meaning a timescale of at minimum a daily cycle, and no more than a few daily cycles." The

10 m reference depth has been used by Holte & Talley (2009) and in general in all MLD estimates based on observations,

except in the Arctic ocean where mixed layers thinner than 10 m are found in summer under sea ice (Peralta-Ferriz &

Woodgate, 2015). In models on the other hand, Griffies et al. (2016) recommend to use the "surface" (that is, the top model

level) as a reference depth. From Fig. 2d, it is easy to see that in spring, changing the reference depth for the density threshold





between 10 m and the surface may cause a large difference in MLD in models where the vertical resolution is sufficiently fine: if a shallow restratification that exceeds the density threshold is present above 10 m, the MLD computed with reference to the

surface can be smaller than 10 m, while the MLD computed with reference to 10 m can reflect the deep winter mixed layer and be more than 100 m deeper. Furthermore, some models may include parameterizations of diurnal cycling (e.g., Large & Caron, 2015) that complicate clear interpretation of the instantaneous sea surface properties but not those at 10 m. The question is whether such differences occur only locally and intermittently, or whether they can affect the time-mean MLD computed in a climate model.

To assess the influence of the reference depth, we have recomputed the MLD using monthly datasets of temperature and salinity from two low resolution OMIP2 models, NCAR-POP and CMCC-NEMO. A potential density threshold of 0.03 kg.m$^{-3}$ and two reference depths (the top model level and 10 m) are used. A monthly climatology of MLDs is then constructed by averaging the years 1989 to 2018. The MLD difference $\delta_{ref}$ between the two reference depths is mapped in Fig. 3 for the months of May (when $\delta_{ref}$ is the largest in the Northern Hemisphere) and September (when $\delta_{ref}$ is the largest in the Southern

Hemisphere). $\delta_{ref}$ is not a noisy field, but instead shows consistent patterns of differences in both models, which are probably related to the shape of the density profiles in different areas of the world ocean. The amplitude of $\delta_{ref}$ is very dependent on the model vertical resolution near the surface (note the different color scale between the models in Fig. 3). The NCAR-POP model has its top level at 5 m depth, close enough to the 10 m reference: $\delta_{ref}$ rarely exceeds 20 m for that model. In contrast, CMCC-NEMO has its top model level at 0.5 m, and $\delta_{ref}$ is larger than 40 m over large areas of the ocean in the climatological average.

Other models confirm this relationship between $\delta_{ref}$ and the near-surface model resolution (not shown): the ACCESS-MOM model with a top level at 1.15 m has a $\delta_{ref}$ similar to the CMCC model, while the IAP-LICOM model with a top model level at 5 m has a $\delta_{ref}$ similar to the NCAR-POP model.



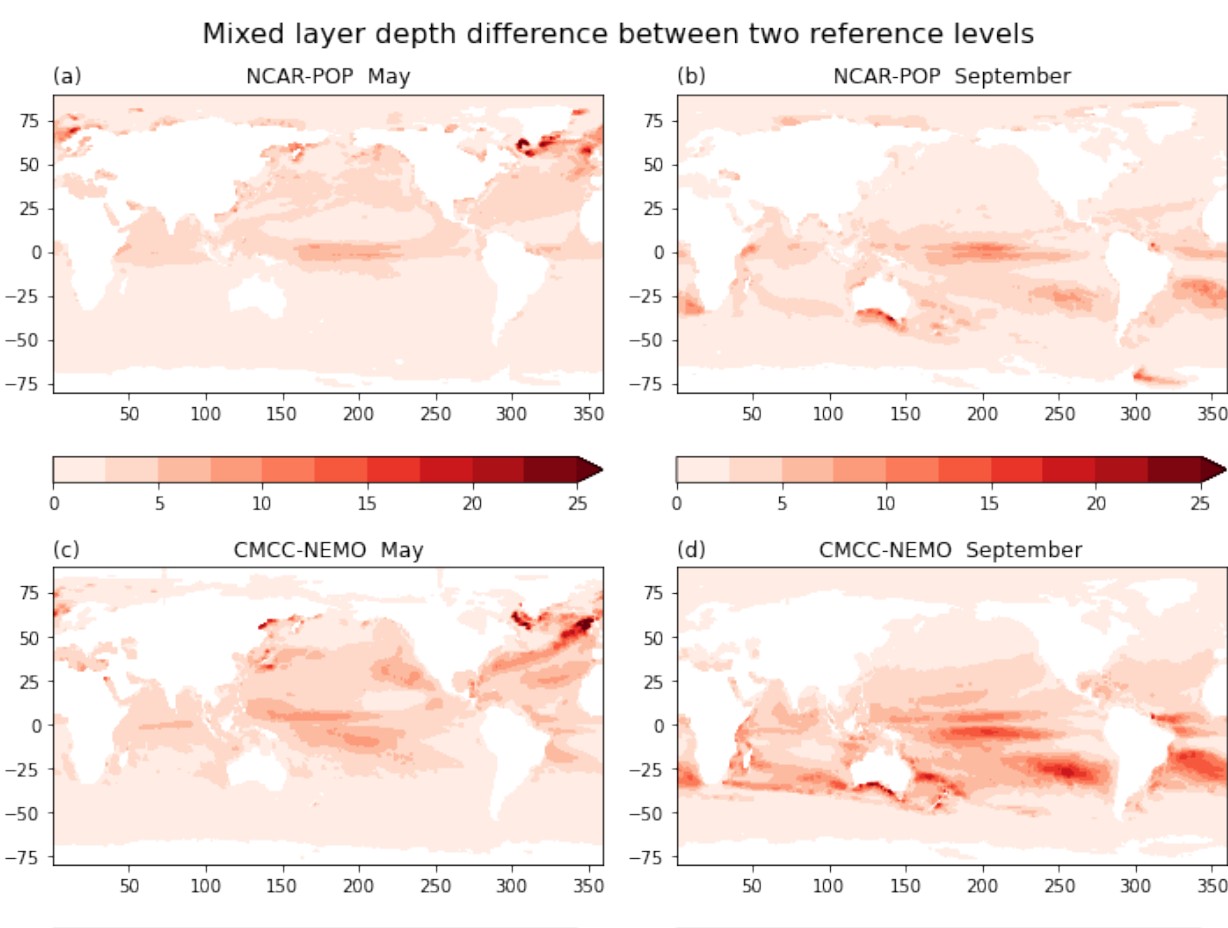

**Figure 3 : Difference (m) between the MLD computed using two reference depths: 10 m and the top model level. The difference is shown for two months and two low resolution OMIP2 models (climatology averaged over 1989-2018).**

In conclusion, using the top model level as a reference depth instead of a reference at 10 m can make a significant and resolution-dependent difference (of the order of 40 m or more) in the monthly climatology of a model MLD when the top model level is close to the surface (about 1m or less). Such differences cannot be ignored, being of the same order of magnitude as inter-model MLD differences found by Tsujino et al. (2020) in some regions and seasons. In this paper, we compute the MLD with a reference depth of 10 m, as advocated by BM04.

### 3.3 Nonlinearity of the MLD revealed by observations

Based on the above discussion, it is clear that in order to compare model MLDs with observational datasets it is necessary to use the same reference level (usually 10 m in profile-based MLD climatologies). However validating MLDs in models is further complicated by the fact that the MLD is a nonlinear function of the density profiles.



This nonlinearity manifests itself in observations, when comparing estimates of MLD based on individual profiles with a MLD computed from climatological profiles; this effect has been documented by BM04. Let us denote a spatio-temporal average by <>; for example, the average over all profiles observed in a 2° box. BM04 compare the spatio-temporal average of MLDs computed from individual profiles, <MLD(p)>, with the MLD computed from the corresponding averaged density profile, MLD(<p>) and show a global map of the relative difference (their Fig. 6). The profile-based average, <MLD(p)>, is generally

deeper by about 25%. BM04 provide a graphic explanation of this result by plotting observed profiles and the two MLD computations in a typical 2° box (their Fig. 7). The fact that <MLD(p)> tends to be systematically deeper than MLD(<p>) is due to the fact that near-surface stratifications, corresponding to shallow MLDs, are generally stronger than the deeper stratifications corresponding to deep MLDs. Although <MLD(p)> is the average of both shallow and deep MLDs computed on individual profiles, the averaged profile <p> has the imprint of the relatively strong near-surface stratification, and thus the

MLD computed on this averaged profile is shallower than <MLD(p)>. Opposite situations are found when the near-surface stratification is intermittent or weak enough so that its imprint on the averaged profile is smaller than the density threshold used to compute the MLD. Such situations are found for example in the subpolar gyres of the Northern Hemisphere in spring (BM04, their Fig. 6).

We expect this consequence of MLD nonlinearity to show up in the comparison of a profile-based climatology of MLD (such

as BM04) with the MLD computed from a gridded climatology of potential density (ISAS; Gaillard et al., 2016). Zonal averages are plotted in Fig. 4 for two seasons. The MLD labeled "deBoyer" is updated from BM04 (de Boyer Montégut, 2022) and has been computed from individual profiles using a fixed density threshold of 0.03 kg.m$^{-3}$ relative to a depth of 10 m, for comparison with OMIP models. The same method has been used to compute the MLD from the monthly ISAS climatology of temperature and salinity. Despite the two products being based on a very similar set of hydrographic observations, both relying

heavily on the ARGO network, there is a systematic difference between the MLDs, with the zonal mean of ISAS MLD (red curve in Fig. 4) being systematically smaller than deBoyer (black curve). This difference is in agreement with the findings of BM04 regarding the average of profile-based MLDs being very often deeper than the MLD of the average profile. It is also in agreement with the findings of Toyoda et al. (2017) who compared the MLD from reanalyses with both profile-based MLDs and MLDs computed from the World Ocean Atlas gridded climatology (their Fig.1).



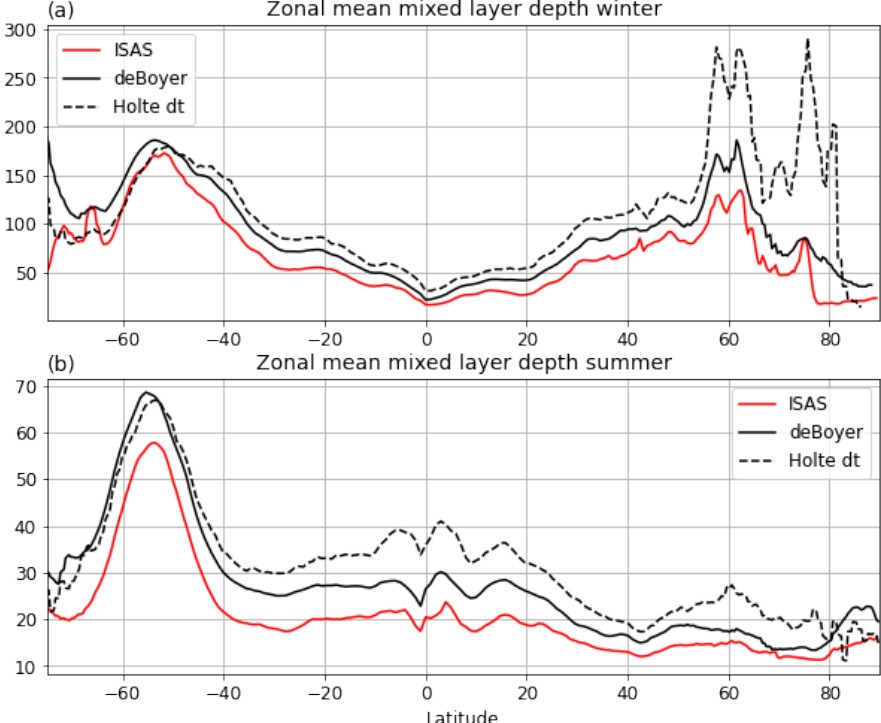

**Figure 4 : Zonal mean MLD (m) as a function of latitude in three different observation datasets (see text for details). The winter (summer) MLD is the average over the months of January to March in the Northern (Southern) Hemisphere, and July to September in the Southern (Northern) Hemisphere, respectively.**

The choice of an observation dataset can influence model evaluations, especially for summer MLDs where differences between datasets are larger relative to the mean MLD. Huang et al. (2014) have evaluated the summer MLDs in 45 CMIP5 low resolution models against the profile-based MLD climatology of BM04. For consistency all of the model MLDs were recomputed from monthly archives with a density threshold of 0.03 kg.m$^{-3}$ relative to a depth of 10 m. They find that the summer MLD is underestimated in the models. In the North Pacific and the North Atlantic, 80% and 82% (respectively) of the CMIP5 models have too shallow mixed layers compared with BM04 (their Figs. 6 and 7). Because the mixed layer computed from a gridded climatology is shallower than a profile-based one, using the ISAS climatology MLD would dramatically reverse their conclusion: only 24% and 27% of the CMIP5 models would underestimate the MLD relative to ISAS in the North Pacific and North Atlantic, respectively. One could argue that the latter result (an overestimation of the summer MLD by the models) is more relevant because the gridded climatology has a spatial resolution similar to the CMIP5 models and eddies are averaged out before the MLD is computed. However, one should note that the difference between the de Boyer and ISAS MLDs is smaller in the Southern Ocean relative to the mean values (averaged between 30°S to 65°S, the summer MLD is 36 m for ISAS and 47 m for deBoyer), so that Huang et al.'s conclusion about CMIP5 models underestimating the Southern Ocean summer mixed layers is valid when using both observational datasets, contrary to the case of the North Pacific and the North Atlantic.





To illustrate how the choice of the MLD reference dataset impact the interpretation of the model results, we have added in Fig. 4 the "threshold mld" provided by Holte et al. (2017); it has been computed from individual profiles using the variable density threshold proposed by BM04 (a density jump equivalent to a temperature difference of 0.2°C at the profile location). This dataset has been used for the evaluation of MLDs in the CMIP models (Fox-Kemper et al., 2021). The threshold in Holte MLD is larger than 0.03 kg.m$^{-3}$ where the SST is larger than 8-9°C, equatorward of 50°S and 50°N, thus the Holte MLD is larger than the de Boyer one (compare the dashed and the solid black curves). The influence of using a variable threshold there is to deepen the mixed layer by about 10 m, which is a relatively large difference in the tropics where the mixed layer is shallow. Southward of 50°S, where the SST is below 8-9°C, the Holte MLD is shallower than de Boyer, because the density threshold is smaller than 0.03 kg.m$^{-3}$. Note that in Fig. 4, the Holte dataset is not comparable with the others north of 50°N because its zonal mean is not computed over the whole ocean area. Holte et al. (2017) provide MLDs binned into 1°x1° grid cells, with no objective analysis nor addition of a climatology to fill up the cells where the number of profiles is not sufficient. The Holte dataset does not include the Arctic, the Greenland Sea, the Baltic, nor any ice-covered region. This is also the case for the more recent GOSML dataset (Johnson & Lyman, 2022). The non-global character of these two datasets make them less suitable for comparison in zonal or large-scale mean with global models, compared with objectively analyzed datasets such as de Boyer Montégut (2022). Note that MLDs computed from climatologies have sometimes been preferred to profile-based MLDs for the evaluation of models; for example, Danabasoglu et al. (2014) and DuVivier et al. (2018) have compared the MLD in forced global models with a MLD computed from the World Ocean Atlas climatology.

**3.4 Influence of the sub-monthly variability**

Figs. 3 and 4 show that the MLD depends on the method used to compute it. Although all the OMIP models used in this paper have computed the MLD online at every time step, the different methods and reference depths used (listed in Table 1) make it difficult to use these online MLDs for intercomparison purposes.

Recognizing this fact, many published model intercomparisons have not used the MLD provided by the modeling centers but rather have recomputed the MLDs from the monthly database of three-dimensional temperature and salinity, in order to use a consistent MLD definition across models (Heuzé, 2021; C. J. Huang et al., 2014). Using a monthly archive means that the submonthly variability of the MLD, driven by the atmospheric synoptic variability and ocean eddies and fronts in high-resolution models, is not sampled. Furthermore, the nonlinearity of the MLD computation may lead to rectified effects, similar to those discussed above, resulting in systematic differences between the MLD computed at high frequency (time step or daily) vs. a MLD computed from monthly density profiles. Toyoda et al. (2017) have found that computation of MLD from monthly datasets of reanalyses underestimates the MLD by 10-20 m in early spring, compared to a computation based on daily datasets. Here we use daily output from two high resolution models to quantify the effect of the sub-monthly variability. The MLD has been computed using the density threshold of 0.03 kg.m$^{-3}$ relative to 10 m depth, for both daily 3D fields (MLD$_d$) and monthly averaged fields (MLD$_m$), for one year. The year 2018 was available for FSU-HYCOM and 1998 for IAP-LICOM. The first point to note is that the mesoscale imprint on the MLD is generally large. This creates a large spatial variability of MLD that





is visible in both MLD$_d$ and MLD$_m$ (Fig. 5). Two regions are selected as examples: the Gulf Stream region in March 2018 (FSU-HYCOM, top panels) and the Southern Ocean in September 1998 (IAP-LICOM, bottom panels). The MLD computed from the monthly mean (right panels) is smoother than the MLD for a particular day of the month (left panel), as expected. However, most of the mesoscale spatial variability is still present in the MLD computed from the monthly mean. Some features

cannot be captured at monthly resolution, such as the thin line of shallow mixed layers from 65°W, 26°N to 57°W, 34°N which is potentially due to precipitation below an atmospheric front.

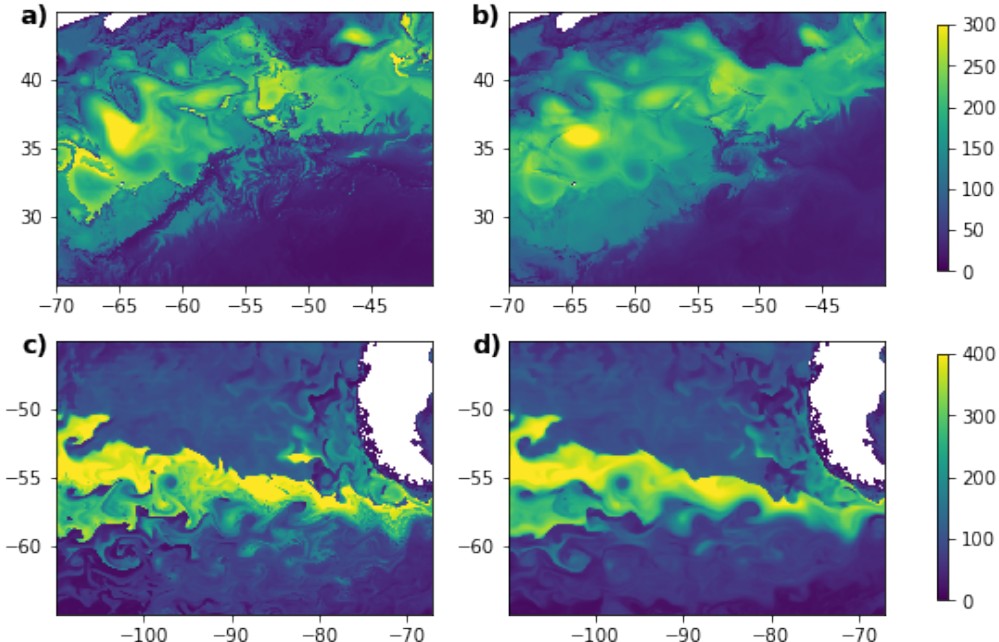

**Figure 5 : Mixed layer depth (m) in the Gulf Stream region in FSU-HYCOM, year 2018 (panels a and b) and in the Southern Ocean west Pacific sector in IAP-LICOM, year 1998 (panels c and d). The left panels a) and c) show the MLD computed from one daily**

**snapshot of temperature and salinity on March 15th. Panels b) and d) show the MLD computed using profiles averaged over the month of March.**

Does the use of monthly density profiles lead to systematic differences in MLD due to the nonlinearity of the MLD computation? We have compared monthly maps of MLD$_d$ and MLD$_m$ (not shown). There are differences exceeding 100 m in some regions and some months, especially in the subpolar North Atlantic and the Southern Ocean in winter and spring. The

zonal average of the difference MLD$_d$ -MLD$_m$ is shown in Fig. 6 for each month. Overall, the difference is positive, consistent with the rectified effect mentioned above. A similar but slightly weaker effect was found by Toyoda et al. (2017) using a low resolution (1°) reanalysis, see their Fig. 2. The effect is very small in summer (about 2 m difference) for both models. It is more important in spring in the northern hemisphere, but in the zonal average, the difference is small relative to the average MLD, and smaller than differences between the different observation-based datasets shown in Fig. 4.





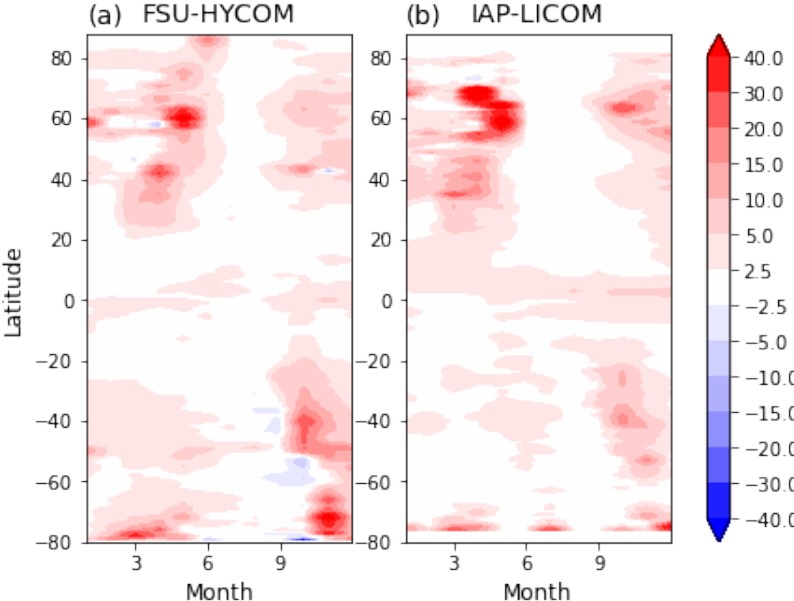


**Figure 6 : Zonal mean of the difference between daily and monthly MLDs (MLDd -MLDm, see text), in meters, averaged for each month. The computation is done in two high resolution models: a) FSU-HYCOM (year 2018) and b) IAP-LICOM (year 1998).**

For our model intercomparison, there is trade-off between using high-frequency online MLDs, but calculated with different methods, vs. recomputing the MLD with a consistent method but a monthly database. Although the loss of submonthly variance

can generate significant rectified effects (of order 100m) in some months at some locations, the zonally averaged difference between $MLD_d$ and $MLD_m$ appears smaller than the differences caused by using different methods to compute MLD, justifying our choice to recompute MLD from monthly temperature and salinity fields whenever possible.

## 4 MLD biases at the global scale

In this section, we compare the climatological MLD biases in the low resolution and high resolution models over the years

1989 to 2018. MLDs are recomputed from monthly archives, using a 0.03 kg.m$^{-3}$ potential density threshold and 10 m reference depth. However, we have kept the MLD computed online for the high resolution 1/16° CMCC model (a MLD referenced to the top model level instead of 10 m), because it has not been possible to re-compute the MLD for that model. The MLD in the high resolution models has been computed on each model native grid. For comparison with observations and with low resolution models, the MLD has been coarsened by spatial averaging to reach a 1° resolution (coarsening by a factor of 10 for

IAP-LICOM, ACCESS-MOM and NCAR-POP, a factor of 12 for FSU-HYCOM and 16 for CMCC-NEMO), before computing further spatial means or other statistics.



### 4.1 Winter mixed layers

Deep mixed layers observed at high latitudes at the end of the winter season receive much attention, because they contribute to forming the water masses that enter the deep ocean and are isolated from the influence of the atmosphere over time scales

of years to centuries. Low resolution models have shown large differences in the amplitude and the location of MLD maxima. This has been demonstrated in the North Atlantic by Danabasoglu et al. (2014) with models forced by the CORE atmospheric forcing (similar to OMIP1), and confirmed by Tsujino et al. (2020) at the global scale with more recent models forced by both OMIP1 and OMIP2 forcings. The winter MLD biases are shown for the six model pairs used in this study (Fig. 7). For the low resolution models, despite using newer versions of the models, the biases in the North Atlantic sector are large and comparable

with the CORE intercomparison of Danabasoglu et al. (2014; see their Fig. 13). FSU-HYCOM overestimates the MLD in the Labrador, Irminger and Greenland seas, as is the case for NCAR-POP but with weaker biases. AWI-FESOM MLD is too large in the Labrador Sea but less biased in the Nordic seas. CMCC-NEMO has more moderate biases in the Labrador and Irminger seas but too deep mixed layers in the Nordic seas. The IAP-LICOM and ACCESS-MOM models were not considered by Danabasoglu et al. (2014), and show rather shallow MLDs in the Labrador Sea while the North-Eastern Atlantic and the Nordic

seas have too deep mixed layers. In the Southern Ocean, NCAR-POP, CMCC-NEMO and ACCESS-MOM show biases similar to the ones documented by Downes et al. (2015, their Fig. 1). FSU-HYCOM has very deep mixed layers close to Antarctica, a feature also noted by Downes et al. (2015). AWI-FESOM has lower biases in the Southern Ocean compared with the other models. IAP-LICOM seems to be an outlier, with overly shallow winter mixed layers in the Southern Ocean.



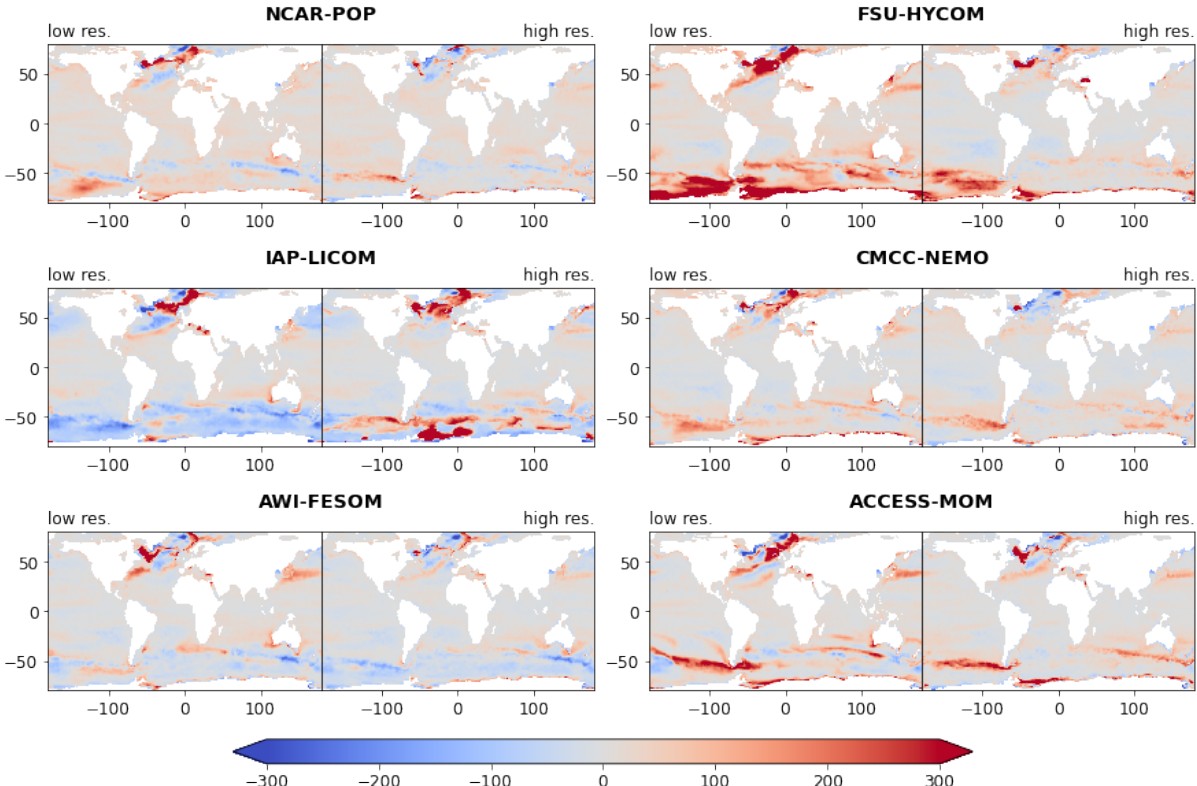

**Figure 7 : MLD biases (m) of models in winter, relative to de Boyer Montégut (2022). For each model, biases are shown for both the low resolution version (left panel) and the high resolution version (right panel). The MLD has been recomputed from monthly archives, with the same potential density threshold and 10 m reference depth as the observations, except for CMCC-NEMO at high resolution where the MLD has been computed online using a reference level of 0.5 m (see text for details). The winter MLD is the average over the months of January to March in the Northern Hemisphere, and July to September in the Southern Hemisphere.**

Tsujino et al. (2020) have used the same low resolution models and some other ones to evaluate the impact of changing the atmospheric forcing from OMIP1 (similar to CORE) to OMIP2. The ensemble mean bias for winter MLD was found to be similar for OMIP1 and OMIP2 (their Fig. 11), suggesting that the biases arise from different model formulations and parameterizations rather than forcing. The different behavior of IAP-LICOM with too shallow mixed layers in the Southern Ocean and the Labrador Sea may be related to the vertical mixing scheme used in this model (see Table 1). This kind of scheme

is usually employed in regional and fine resolution models, with a horizontal resolution of less than 1 km and time-steps on the order of minutes. When it is applied in a climate model with coarse resolution and large time-step, the timescale of turbulent kinetic energy variability is not resolved (Reichl & Hallberg, 2018). Therefore, the related diffusivity may be significantly underestimated. Further tuning of its parameters may attenuate errors in MLDs.

MLD biases are generally reduced in the high resolution models (Fig. 7). Low resolution models tend to exhibit deep biases

in some eddy-rich regions (Gulf Stream and Kuroshio extension, Agulhas basin) that seem reduced at high resolution, possibly pointing out a systematic effect of eddies on the restratification of the mixed layer. However, the amplitude and pattern of the



changes are model-dependent. For example, the deep mixed layer bias in the Kuroshio extension region is clearly reduced in AWI-FESOM, ACCESS-MOM and FSU-HYCOM at high resolution, but the other three model pairs have a lower bias at both resolutions in that region. In the Antarctic Circumpolar Current both deep and shallow biases are present at both resolutions.

In the South-East Pacific sector near 50°S, the shallow bias at low resolution in NCAR-POP and IAP-LICOM is replaced by a deep bias at high resolution; in contrast, the deep bias in ACCESS-MOM is reduced. This suggests that the MLD varies due to changes in the circulation and water masses between the different resolutions, and not only due to the explicit representation of mesoscale eddies. One salient feature in Fig. 7 is the excessive convection that develops at high resolution in the Weddell Sea and in the Labrador Sea in the IAP-LICOM model. It is well known that although the Antarctic Bottom Water (AABW)

is formed on the shelves and subsequently sinks to the bottom along the slope of the Antarctic continent, many climate models form AABW by open-ocean convection in the Weddell gyre, and display unrealistic deep mixed layers there (Griffies et al., 2009; Heuzé, 2021). This loss of stratification in the Weddell gyre was thought to be dependent on eddy parameterizations in low resolution models (Griffies et al., 2009). The fact that this problem appears in IAP-LICOM with resolved eddies will need further investigation.

**4.2 Summer mixed layers**

At mid and high latitudes, a strong near-surface stratification is created in summer due to positive surface heat flux, making the mixed layer shallow. Relatively shallow mixed layers dominate the tropical latitudes all year round (Fig. 4), so that the annual average of the mixed layer in the world ocean, computed from the de Boyer Montégut (2022) dataset, is only 53.4 m. Model biases are accordingly smaller in summer (Fig. 8) than in winter (Fig. 7). For low resolution models, the main features

that stand out in Fig. 8 are a tendency for too shallow summer mixed layers in the Southern Ocean, and a band of too deep mixed layers around 5°N: this is similar to the multi-model bias shown - but not discussed- by Tsujino et al. (2020, their Fig. 12b). The Southern Ocean shallow bias is a longstanding problem that was pointed out by Griffies et al. (2009), who noted that in low resolution models the bias was dependent on the lateral mixing scheme and the details of the implementation of the Gent-McWilliams parameterisation in the surface layers. The bias is reduced but does not disappear in most of the high

resolution models, pointing to model deficiencies other than the parameterization of eddies. The shallow bias in the Southern Ocean is the largest in IAP-LICOM at both low and high resolution, suggesting again a role of the vertical mixing scheme. Other biases are more difficult to interpret because there are differences other than horizontal resolution between the two members of each model pair. In the case of NCAR-POP, the shallow bias is stronger in the high resolution model; this may be due to the lack of a parameterization of Langmuir-induced mixing (a parameterization that was present in the low resolution

version; Table 1). In the case of CMCC-NEMO, the vertical mixing is exactly the same at both resolutions, but the mixed layer is too deep in the eddy resolving simulation. The atmospheric forcing being the same, this deep bias must be due to changes in the circulation and the vertical profiles of temperature and salinity. A full analysis of such biases, similar to DuVivier et al. (2018) or Small et al. (2021) is beyond the scope of our paper. We hypothesise that changes in the stratification may result



from the use either of different vertical resolutions or two distinct sea ice models in the two members of the CMCC-NEMO

pair, the latter resulting in large differences in sea ice concentration.

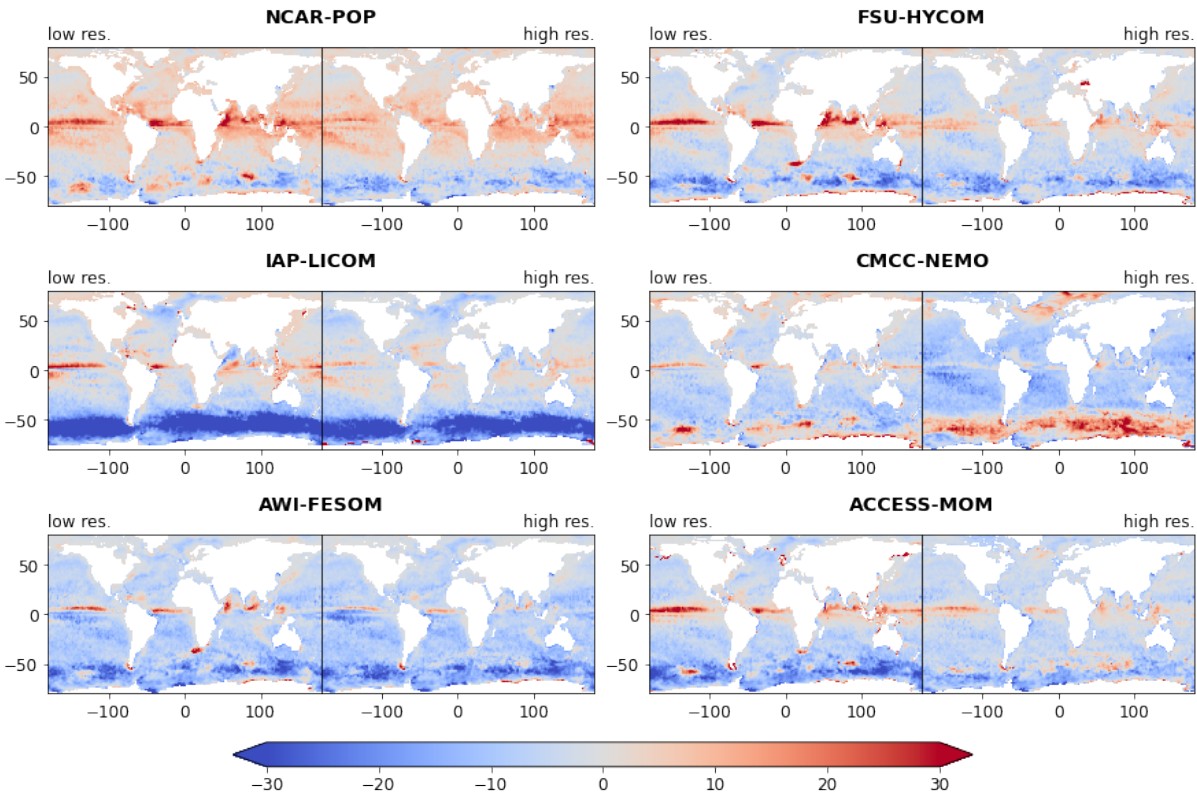

**Figure 8 : MLD biases (m) of models in summer, relative to de Boyer Montégut (2022). For each model, biases are shown for both the low resolution version (left panel) and the high resolution version (right panel). The MLD has been recomputed from monthly archives, with the same potential density threshold and 10 m reference depth as the observations, except for CMCC-NEMO at high**
**resolution where the MLD has been computed online using a reference level of 0.5m (see text for details). The summer MLD is the average over the months of January to March in the Southern Hemisphere, and July to September in the Northern Hemisphere.**

The other region of strong biases is the latitude band near 5°N. As shown in Figs. 1 and 4, the observed MLD has a local

maximum there between two minima, one at the equator due to equatorial upwelling and one near 10°N which marks the

position of the Intertropical Convergence Zone (ITCZ) during the Northern Hemisphere summer. This local maximum at 5°N

is overestimated to various degrees in the low resolution models (Fig. 8). It is quite remarkable to note that this bias is reduced

at high resolution in all the models. The amplitude of the bias at low resolution does not seem related in a straightforward

manner to model parameters such as the vertical resolution (similar for ACCESS-MOM and CMCC-NEMO), the meridional

resolution near the equator (lower in IAP-LICOM compared to NCAR-POP), or the vertical mixing schemes. This region is

characterized by a salinity control of the stratification at the base of the mixed layer (Helber et al., 2012, their Fig. 15c). An

examination of profiles in the Central Pacific suggests that the low resolution models with the strongest deep bias underestimate





the salinity increase at the mixed layer base (not shown). The improvement of the MLD at high resolution may be due to both advection of salty water masses by the complex near-equatorial current system and to the lateral mixing by eddies and waves such as tropical instability waves. Similar challenges regarding modeling deep salinity maxima and their influence on the MLD, even at high resolution, have been pointed out by duVivier et al. (2018), in the case of the "deep mixing band" of the
Southern Ocean.

### 4.3 Influence of model resolution on the zonal mean biases

Fig. 9 (a,b) gives an overview of the zonal mean MLD biases. The differences across models, represented by color shading, are larger than typical differences between observation-based datasets (ISAS and deBoyer are indicated in black lines, as examples). The range of biases is reduced in the eddy resolving models compared with the low resolution ones, but biases
remain significant. We have demonstrated in the previous section that the spatially-averaged MLD is usually larger when it is computed from density profiles sampled at high resolution (as in a profile-based climatology) compared with a computation on spatially averaged fields of density (a gridded climatology).

Could the sampling effect cause a systematic difference of MLDs in high and low resolution models? In that case, one would expect the difference between high resolution and low resolution MLDs to have some similarity with the difference between
the profile-based MLD climatology of de Boyer and the MLD computed from the ISAS climatology. Fig. 9 (c,d) shows that it is not the case. In winter (c), the MLD differences between high and low resolution models are much larger than the differences between the two observation datasets. The MLDs at high resolution are shallower for most models and in most latitude bands (the high-low difference in Fig. 9 is negative, contrary to the positive difference between the two observational products). In summer (d), the high-low resolution differences are also mostly negative north of 40°S, of opposite sign and smaller than the
difference between the observational datasets. South of 40°S, three models (CMCC-NEMO, IAP-LICOM and ACCESS-MOM) have a deeper mixed layer at high resolution, while the other three models have shallower mixed layers. Thus, the MLD changes brought by the high resolution do not result from higher spatial sampling only. As mentioned above, the differences could have a dynamical origin (explicit representation of eddies at high resolution), but different parameterizations or different vertical grids for the two members of each model pair also contribute (Table 1).




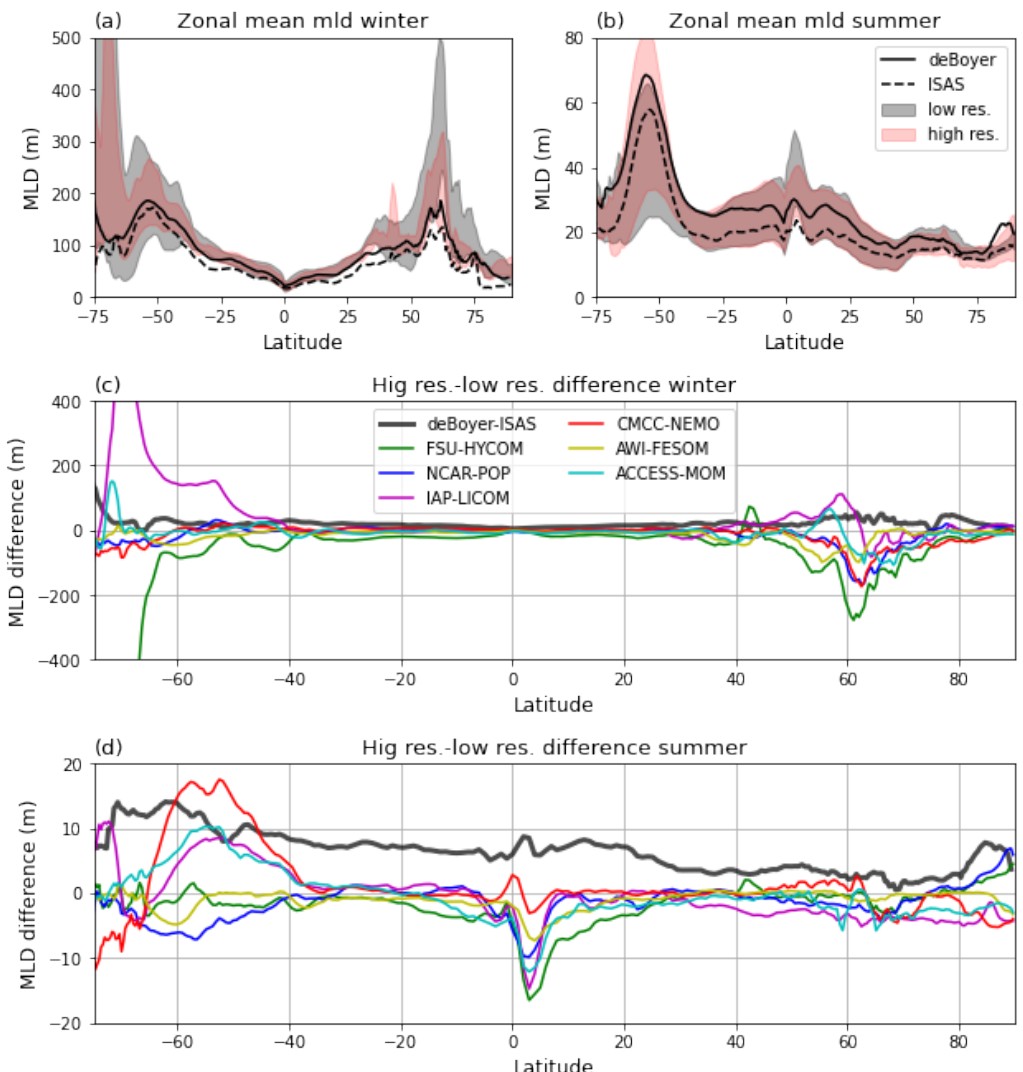

**Figure 9 : Zonal mean MLD biases and their dependency on model resolution. Top panels: zonal mean biases for two observation datasets (black curves) and for the range of low resolution models (grey shading) and high resolution models (red shading), for the winter season (a) and the summer season (b). The lower panels show the zonal mean of the difference between the MLD of the high resolution models and the low resolution models, for each model pair, for winter (c) and summer (d). The dark grey curve in panels**
**(c,d) represents the zonal mean of the difference between the profile-based MLD climatology of de Boyer Montegut ( 2022) and the MLD computed from the ISAS climatology. For the CMCC model, the online MLD computation is used for both the high resolution and the low resolution model, for consistency.**

## 5  MLD biases in water mass formation regions

In this section, we consider in more detail the seasonal cycle and the maximum MLDs in regions of deep and intermediate

water mass formation. Mesoscale and submesoscale dynamics play important roles in these regions, from preconditioning



before convection events (Lherminier et al., 1999) to restratification after convection (Gelderloos et al., 2011). In the Northern Hemisphere, we define three regions, the Labrador Sea, Irminger Sea, and Greenland Sea, based on the map of maximum MLD in the de Boyer (2022) climatology (Fig. 10). In numerical simulations, convection may occur outside these areas, but here we focus on the way models (especially the high resolution ones) have the capability to form water masses at the right

location. Note that in these weakly stratified cold seas, the MLD based on the 0.03 kg.m$^{-3}$ density threshold is larger than MLDs based on other criteria (Courtois et al., 2017; BM04). Fig. 10 compares the seasonal cycle of simulated MLDs with observations. In all regions, models capture the asymmetry of the seasonal cycle, with a slow deepening of the MLD in autumn and a rapid shallowing in spring. In the Greenland Sea, all the low resolution models underestimate the winter MLD (grey shading). In fact, as shown in Fig. 7, these models have strong positive MLD biases all along the warm and salty Norwegian

current, resulting in deeper mixed layers in the Norwegian Sea than in the Greenland Sea, contrary to observations. There is a marked reduction of these biases at high resolution (Figs. 7 and 10) for three models: NCAR-POP, FSU-HYCOM and ACCESS-MOM. In the Labrador Sea, low resolution models show a range of shallow and deep biases, but all the high resolution models show deeper MLDs than the observations. In the Irminger Sea, almost all the low resolution models tend to overestimate the MLD (Fig. 10, grey shading), while high resolution models have biases of either sign. In these three regions

where deep convection occurs, high resolution does not bring improvement uniformly across regions and models. There is no systematic reduction in uncertainties: in the Greenland Sea and the Irminger Sea, the spread of maximum MLD is larger across high resolution models, compared to low resolution ones.





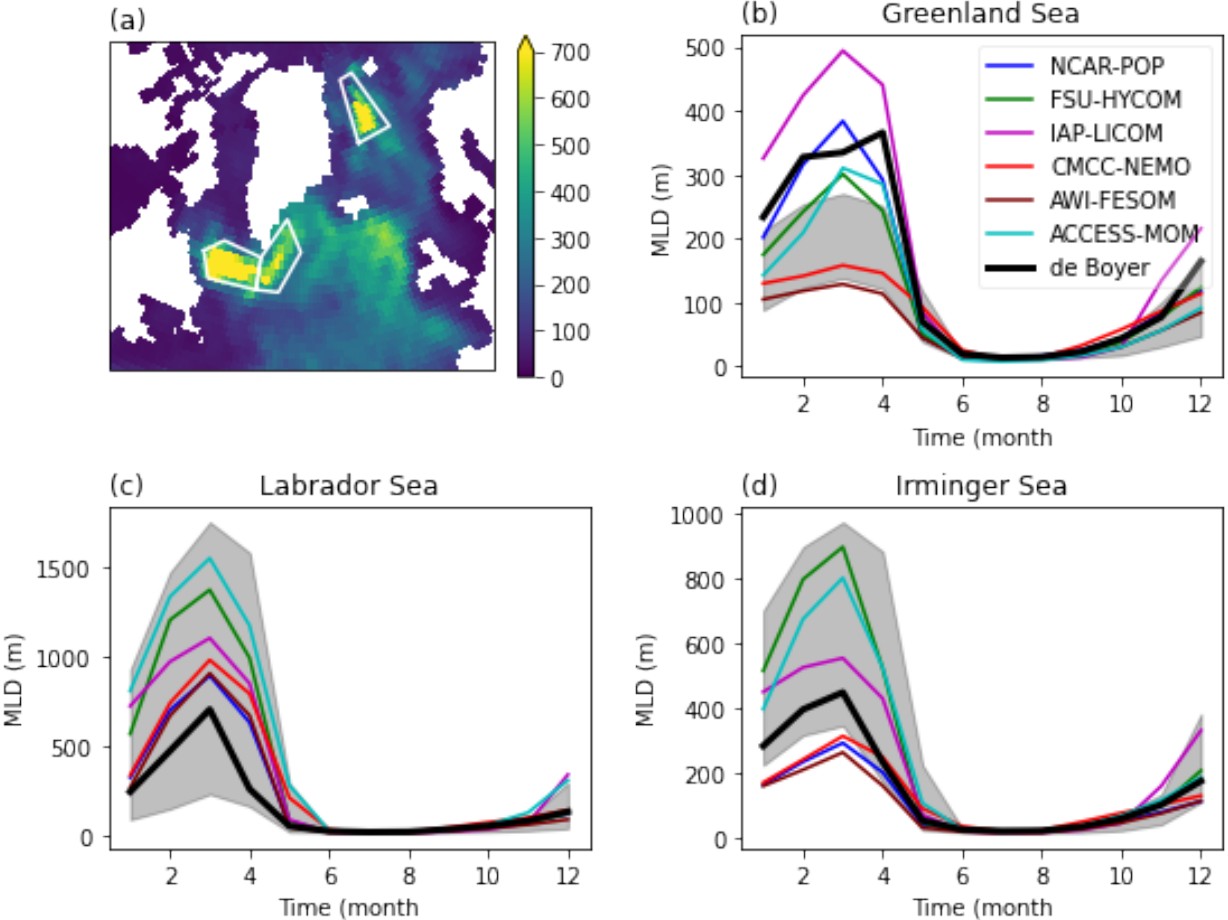

**Figure 10 : Seasonal cycle of MLD in three regions. (a) map of the maximum MLD (de Boyer Montégut, 2022) indicating the three**
**regions of deep water formation (Labrador Sea, Irminger Sea, and Greenland Sea). In panels (b-d), the seasonal cycle of the**
**observed MLD (m) averaged over each region is plotted in black and the high resolution models in color. The time axis is in months.**
**The grey area is the range of the low resolution model MLDs.**

Let us consider regions of intermediate or mode water formation, outlined in Fig. 11. These regions are also regions of active

mesoscale turbulence, especially the region of 18°C water formation south of the Gulf Stream, the Kuroshio region, and the

band of deep MLDs on the northern flank of the Antarctic Circumpolar Current in the West Pacific and South Australian

sectors, where SubAntarctic Mode Waters (SAMW) are formed. We also consider the East Subpolar Atlantic, a region where

eddy activity is less intense but which plays an important part in the formation of North Atlantic subpolar mode waters and

preconditioning of deep convection in the Labrador and Irminger Seas (Courtois et al., 2017). The high resolution models have

been shown to represent well the eddy variability compared with satellite observations (Chassignet et al., 2020; Iovino et al.,

2016). A substantial improvement is found in the seasonal cycle of MLD at high resolution in the three regions of the Northern

hemisphere. The spread of the models is much reduced compared to low resolution models, which suggest that intermediate




water mass formation mechanisms could be much more robust at high resolution. For example, the excessively deep MLDs that occurred in the low resolution models in the Eastern subpolar Atlantic disappear at high resolution. On the other hand, there is no improvement at high resolution in the two Southern Ocean regions, which are part of the "deep mixing band"

analyzed by DuVivier et al. (2018) using ARGO observations and the NCAR-POP model at low and high resolution. Their simulations use the CORE (OMIP1) forcing, rather than the JRA55-do reanalysis of OMIP2, but overall they document biases very similar to ours, which is expected considering the robustness of winter MLD to the forcing (OMIP1 or OMIP2) demonstrated by Tsujino et al. (2020). DuVivier et al. (2018) find a relationship between the deep mixing band and the stratification, and argue that a different representation of the subsurface salinity minimum is the cause of the difference between

the two NCAR-POP models. The salinity minimum has a more realistic strength in the high resolution POP, due to the explicit representation of eddies and a more realistic circulation (the Agulhas retroflexion, for example), but the salinity maximum is too deep which causes the winter MLD to be overestimated (DuVivier et al., 2018). The complex interplay between circulation, eddies and water mass distribution in the Southern Ocean is clearly a challenge for the models considered here, even at high resolution.






**Figure 11 : Seasonal cycle of MLD in five regions. (a) map of the maximum MLD (de Boyer Montégut, 2022) indicating the five regions of intermediate water formation  (East subpolar Atlantic, Gulf Stream, Kuroshio, South Pacific and South Australia). In panels (b-f) the seasonal cycle of the observed MLD (m) averaged over each region is plotted in black and the high resolution models in color. The time axis is in months. The grey area is the range of the low resolution model MLDs.**



**6 - Conclusions**

Increasing the horizontal resolution in ocean models often produces mixed results, rather than a universal improvement of the model solution (Chassignet et al., 2020). The same is true in our investigation of the MLD in the OMIP models. While the MLD biases are generally reduced by refining the horizontal grid, spurious deep winter mixed layers are found in the high resolution models at some locations, for example in the Weddell Sea for IAP-LICOM and in the Labrador Sea for ACCESS-

MOM. It is encouraging to see that when eddies are resolved at high resolution, the inter-model differences are reduced in some eddy-rich regions (Gulf Stream and Kuroshio). Unfortunately, it is not the case in the Antarctic Circumpolar Current nor in the subpolar regions: increasing the spatial resolution does not improve the robustness of the model solution uniformly over the world ocean. MLD biases differ more across models than across resolution. The different biases come from differences in model formulations, vertical resolution, and parameterizations. Perhaps because of its nonlinear character, the MLD is a

variable that is not well constrained by the imposed OMIP atmospheric forcing, contrary to the sea surface temperature (SST). The inter-model difference in SST in OMIP is relatively small (Tsujino et al., 2020) compared with SST differences across CMIP coupled models such as displayed by Eyring et al. (2021, their Fig. 3.3). On the other hand, the inter-model differences of MLD in our model pairs are similar to those found in CMIP6 models (Heuzé, 2021).

The zonal average of the MLD difference between the high and low resolution models, displayed in Fig. 9, shows that the

MLD is generally shallower in high resolution models. This difference cannot be due to a rectified effect of the sampling, which would make the mixed layer deeper at high resolution, as pointed out by BM04 and demonstrated in section 3. The zonal mean, global view suggests that the role of mesoscale eddies in restratifying the mixed layer is a dominant factor to explain the dependency of MLD on horizontal resolution, and that this effect of eddies is not parameterized adequately in the low resolution models. Note that by considering the spatial modulation of the MLD by eddies in observations, Gaube et al.

(2019) reached the opposite conclusion. They argue that the MLD is globally deeper due to the presence of eddies. There is no contradiction here, because evaluating the contrast between MLD inside and outside the eddies, as done by Gaube et al. (2019), does not reveal the dynamical role of eddies on the stratification. The simplicity of the zonal mean view, however, may be misleading because it masks compensating biases of both signs. MLD biases depend not only on the explicit representation of eddies but also on the circulation in relation to water masses that control the stratification at the mixed layer

base. This is especially true in the Southern Ocean.

Regarding the importance of eddies for restratification, our results are still limited by the fact that even our high resolution models do not represent submesoscale dynamics. Submesoscale motions are more efficient than mesoscales for restratifying the mixed layer  (e.g., Fox-Kemper et al., 2008; Mensa et al., 2013), so that the shoaling of the MLD with resolution could be even larger if higher resolution models were considered. However, our model intercomparison does not suggest a dominant

role of the parameterization of submesoscales, as models in this ensemble with and without the Fox-Kemper parameterizations display similar biases. Thus, other drivers of errors such as numerics, mixing parameterizations, vertical resolution or the impact of the different definitions for modeled MLD (monthly archive of horizontally gridded values) and observed MLD



(from profiles localized in space and time), exceed the magnitude of impact of these parameterizations in many regions in these models.

The eddy-rich models considered here are promising tools to study mixed layer statistics. Johnson and Lyman (2022) have recently published the GOSML dataset of MLD statistics based on ARGO data. They find that the distribution of MLD is non gaussian, with large skewness and kurtosis that vary seasonally and spatially. The MLD variance also displays seasonal variations, and depends on the MLD itself (regions with large MLDs have a large MLD variance). We have found similar properties of the MLD statistics in the high resolution models, but the MLD variance computed from monthly averages is low

compared with the observations. The MLD varies at high frequencies due to diurnal and synoptic atmospheric forcing (Whitt et al., 2019), which means that archives of MLD at high frequency will be needed to validate higher order statistics of the mixed layer in eddy-rich models. The stratification at the base of the mixed layer is also an important feature that needs to be well simulated in models. Serazin et al. (2023) show that this density gradient, the upper ocean pycnocline, has a small thickness (a median value of 23 m globally). The increased vertical resolution of some eddy-rich models (98 levels for CMCC-

NEMO, for example) could bring improvements in the representation of processes at the mixed layer base.

An important conclusion of our work is that the protocol for computing the MLD in OMIP and CMIP (Griffies et al., 2016) needs revision. Firstly, the density jump should be computed with reference to a depth of 10 m, not with the top model level, because the vertical grids differ in different models and we have demonstrated that in some seasons a difference of less than 10 m in the reference depth can lead to more than 40 m difference in the MLD climatology. Secondly, all models should use

the density threshold of 0.03 kg.m$^{-3}$ (or the corresponding buoyancy threshold, Griffies et al., 2016) to facilitate intercomparisons, and these models should be evaluated against observational products formulated consistently with this definition (e.g., de Boyer Montégut, 2022). Finally, this variable should be stored at high frequency (hourly or daily) and calculated online for averages, variances, or other statistics retained only occasionally. For models with very high vertical resolution near the surface (such as CMCC-NEMO and ACCESS-MOM) the computation of the MLD relative to the top

model level would be an interesting additional diagnostic at hourly frequency, as it would resolve the diurnal cycle of the MLD and allow assessment of its rectified impact at longer time scales, but different observations than the profile-based ones used here would be needed to evaluate this metric.

**Code and data availability**

The dataset produced by de Boyer Montégut (2022) is published on the SEANOE repository. The following OMIP model output, published on the Earth System Grid Federation, has been used: ACCESS-OM2 (Hayashida et al., 2021), CESM2 (Danabasoglu, 2019), CMCC-CM2-SR5 (Fogli et al., 2020), FGOALS-f3-H (Lin, 2020), FGOALS-f3-L (Lin, 2019). The 0.1° ACCESS-MOM data is available from http://dx.doi.org/10.25914/608097cb3433f. The data used to produce the figures and

the corresponding python notebook are archived on Zenodo, doi:10.5281/zenodo.7656425.



**Author contribution**

AMT coordinated the conceptualization, carried out the investigation and wrote the first draft of the manuscript. EC, AB, XX, AH, DI, AK, YL, PL, HL, DS, QW, SY provided model-based datasets and expertise for the interpretation of diagnostics. 635 CBM prepared the dataset used to validate the models. AMT, BFK, CBM, EC, AH, DI, AK, JLS, HL, GS, DS, QW, SY participated in the conceptualisation, edited and reviewed the manuscript.

**Competing interests**

The authors declare that they have no conflict of interest.

**Acknowledgments**.

We thank the present and past members of the CLIVAR Ocean Model Development Panel who have designed and supported the Ocean Model Intercomparison Project. This work is a contribution to the MixED Layer hEterogeneitY (MEDLEY) project. MEDLEY has received funding from JPI Climate and JPI Oceans under the 2019 Joint Call, managed by the French Agence Nationale de la Recherche, under contract 19-JPOC-0001-01. The contributions of BFK are supported by NOAA 645  NA19OAR4310366 and NSF 2148945. The contribution of LYW, LPF and LHL are supported by NSFC 41931183. The contributions of AEK are supported by ARC LP160100073 and the Australian Government's Australian Antarctic Science Grant Program. The COSIMA consortium (www.cosima.org.au) provides the ACCESS-OM2 model suite using resources of the National Computational Infrastructure, which is supported by the Australian Government.

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
