# Peer review of "The Mixed Layer Depth in the Ocean Model Intercomparison Project (OMIP): Impact of Resolving Mesoscale Eddies"

_EGUsphere, 2023_

## Referee Comment (RC2)

A review of egusphere-2023-310 :

**The Mixed Layer Depth in the Ocean Model Intercomparison Project (OMIP):**

**Impact of Resolving Mesoscale Eddies**

*by Anne-Marie Treguier et al.*

*Summary*

The aim of this MS is to estimate the effect of resolved mesoscale processes on the ocean mixed-layer depth (MLD). To do this, the authors have diagnosed the ocean mixed-layer depths in a set of similarly-forced runs of ocean models submitted to phase 2 of the Ocean Model Intercomparison Project (OMIP2) available at both eddy resolving lateral resolution (~0.1°) and coarse CMIP-style 1° resolution. They are then compared with the observation-based dataset of de Boyer Montegut et al. (2022); henceforth BM22, an update of de Boyer Montegut et al. (2004); henceforth BM04.

The authors first discuss how best to define mixed-layer depth. This is always an ambiguous concept, but they follow BM04 in trying to capture the depth over which "surface fluxes have been recently mixed ... meaning a timescale of at minimum a daily cycle, and no more than a few daily cycles."

This depth is essentially the nighttime MLD, and they again follow BMO4 in taking the MLD to be the shallowest depth where the density $\rho$ is 0.03 kg m$^{-3}$ greater than $\rho_{10m}$, that at 10m, i.e. $\rho - \rho_{10m} \geq 0.03$kg m$^{-3}$, so as to exclude shallow (1–3 m) daytime stratification. They follow Griffies at al. (2016) (G16) in using this simple density criterion rather than the density difference implied by a temperature difference of 0.2° at the surface (which depends on the thermal expansion coefficient and hence the surface temperature and salinity) that is used by e.g. Holte et al. (2017). Monthly-mean outputs for May and September from the NEMO coarse-resolution model that had fine vertical resolution (~1 m near the surface) showed that indeed taking the density difference from that at the surface (uppermost grid level); i.e. $\rho - \rho_1 \geq 0.03$ kg m$^{-3}$ rather than $\rho - \rho_{10m} \geq 0.03$ kg m$^{-3}$ gave MLD that were widely up to 40m shallower. For the NCAR-POP model with 5 m vertical resolution that did not resolve the diurnal cycle properly, differences were less marked.

They then discuss the well-known importance of using spatial averages of MLD calculated from individual profiles rather than calculating MLD from the spatially-averaged climatology of temperature and salinity. By comparing the BM22 climatology calculated from individual profiles with the a MLD field calculated from the recent ISAS temperature and salinity climatology they show that the latter underestimates the zonally-averaged MLD by typically 5 m in summer and 25 m in winter.

Next they investigate the impact of non-linearity in the time averaging. In both high and low resolution versions of a couple of the submitted model-pair runs, the monthly average of

the online-averaged MLD calculated every time step was generally shallower than the MLD calculated from the monthly-average temperature and salinity: by about 2–5 m in the summer and by 20–40 m in winter. These differences, though significant, were generally not as great as those arising from using the surface-layer instead of the 10-m density.

Motivated by the above they settle on a definition of the model MLD as the shallowest depth where as $\rho - \rho_{10m} \geq 0.03$ kg m$^{-3}$, calculating the density from the monthly-average temperature and salinity: this definition has the advantage that it can be calculated for all models in the same way, rather than using the online-averaged MLD that are defined using different criteria in different models.

They than address the question of the impact of resolution on MLD by examining the summer and winter MLD defined as above for the various models both at eddy-resolving and coarse resolution, and comparing against the observational dataset of dBM22.

The IAP-LICOM model pair is a clear outlier: at both coarse and high resolution it has far too shallow (by >30 m) summer MLDs in the ACC, and at high resolution way too deep ( by >200–300 m) winter MLDs in the Nordic Seas and parts of the ACC. The too-shallow MLDs are ascribed to the use of the Canuto ML model that is normally used with short time steps (a few minutes) in models with longer time steps (up to ~1 hr). The FSU-HYCOM model has too deep ( by >200–300 m) winter MLDs in the ACC at both coarse and fine resolution; the reasons for this are unclear.

The other models (ACCESS-MOM, CNCC-NEMO, NCAR-POP and AWI-FESOM) all tend to have excessively deep winter MLDs over the Labrador and Nordic Seas (and over parts of the ACC) at coarse resolution, but this deep bias is considerably reduced at high resolution. Summer MLD biases seem to be generally less sensitive to resolution, though the excessive near-equatorial MLDs produced at coarse resolution by all four models (and FSU-HYCOM) are shallower at high resolution. It is suggested that the better resolved near-equatorial flow at high resolution transports salty water masses more correctly, giving more realistic vertical stratification. CMCC-NEMO does however give deeper summer ML over the ACC at high resolution than at coarse resolution; this may be associated with the coarser vertical grid used at low resolution.

The performance of the models in simulating the annual cycle of MLD over key deep-water formation sites (the Greenland, Labrador and Irminger Seas) is then considered, focusing on the high-resolution runs, since they perform better than the coarse runs in these regions. Wide model to model variation (e.g. in the Labrador Sea ACCESS-MOM MLD > 1500 m while NCAR-POP MLD ~800 m) is evident in the maximum MLDs in late winter/early spring. NCAR-POP perhaps performs the best. MLDs in intermediate water generation sites are then considered. The models perform very well in the eastern sub-polar Atlantic, and fairly well in the Sargasso Sea and Kuroshio, with relatively little model spread but with a consistent deep bias of ~50 m, though ACCESS-MOM is a deeper outlier (~30 m deeper). However over both the ACC south of Australia and in the ACC in the eastern South Pacific, there is considerable model spread, with ACCESS-MOM again a deeper outlier.

Conclusions are then:

1.  Model MLD should be defined as the shallowest depth where as $\rho - \rho_{10m} \geq 0.03$ kg m$^{-3}$ and then compared against consistently calculated datasets like dBM22.

2.  Model MLD should be calculated online and then statistical moments such as averages and standard deviations calculated over longer periods.

3.  The impact of the diurnal cycle could and should be investigated in models with sufficient vertical resolution (~1 m near the surface) by considering the density difference from the top level $\rho - \rho_1 \geq 0.03$ kg m$^{-3}$.

4.  Finer lateral resolution mostly gives shallower, generally more realistic, winter MLDs.

5.  Finer lateral resolution does not always lead to more consistent results between models (e.g. over the ACC) , though more consistent maximum winter MLDs are seen in the eastern subpolar Atlantic, the Sargasso Sea and Kuroshio.

*Major comments*

This is an important and timely piece of work on a key topic that is only going to become more important for climate science. The work presented here is rigorous and careful, and the authors have taken care to point out the inconsistencies in the model datasets employed here —there are not many eddy resolving runs available forced by the same atmospheric forcing datasets, and different vertical resolutions and parameterisations are used by different models. Hopefully this MS will stimulate further work on this subject.

The suggested definition of the model MLD as the shallowest depth where as $\rho - \rho_{10m} \geq 0.03$ kg m$^{-3}$ (at least when calculating MLD online) should be followed in future work.

It is an important finding that finding that (generally) the MLD thus defined is indeed shallower at eddy-resolving resolution, presumably because the eddy parameterisations at coarse resolutions are not as effective at fluxing buoyancy upwards as are the resolved eddies at coarse resolution.

It is also interesting to see how sensitive the MLD are to the actual ML model and to model vertical resolution, especially in summer.

I was not fully convinced, however, of the argument that the MLD calculated from the monthly-average temperature and salinity, $MLD(\langle T \rangle, \langle S \rangle)$, was a totally adequate reflection of the monthly-mean of the MLD calculated online at each time step $\langle MLD(T, S) \rangle$, and thus a fully comparable field to the MLD calculated from individual observational profiles. For the zonally averaged $\langle MLD(T, S) \rangle - MLD(\langle T \rangle, \langle S \rangle)$ in Fig. 6 seem to reach ~30–40 m in late spring. Moreover of the two models plotted in Fig. 6, IAP-LICOM is very much an outlier, and FSU-HYCOM has notably coarse vertical resolution (~ 36 layers) and so would not expect to show much of an effect, especially in summer. Fig. 6 would be more convincing if it included output from a higher resolution, "good" model, like CMCC-NEMO which has 98 levels. Why

not use NCAR-POP and CMCC-NEMO as were used in Fig. 3 to show the impact of using different reference levels?

I agree that for purposes of comparability it was necessary to use MLD($\langle T \rangle, \langle S \rangle$), but suggest that it would have been better to have tried a larger density difference criterion when using monthly-mean averages. For it is conventional when calculating MLD using monthly-mean observational climatologies to use a larger density difference, e.g. WOA 2018, which uses $\rho - \rho_{10\text{m}} \geq 0.125$ kg m$^{-3}$ s. It would be relatively straightforward to calculate MLD using monthly-mean $T$ and $S$, with a range of density differences starting up from 0.03 kg m$^{-3}$ and ranging up to 0.125 kg m$^{-3}$ using e.g. CMCC-NEMO, and choose the density difference that minimised deviations from the monthly-average of the MLD calculated online with the 0.03 kg m$^{-3}$ density criterion (i.e minimise the difference field plotted out in the panels in Fig. 6).

*Recommendation*

This manuscript will be very useful for the field and should be published subject to minor corrections.

*Detailed comments*

P6, l138. Does IAP-LICOM include the wave generated mixing of Qiao, et al., GRL, 2004?

P6, l164 "NEMO version 3.64" Is this what you mean?

P14, Fig. 4. Caption. Observation datasets ⇒ observational datasets.

P17, Fig. 4. Why not use NCAR-POP and CMCC-NEMO for consistency?

George Nurser

---

## Author Comment (AC1)

egusphere-2023-310

Reply to review comment #2 by George Nurser

We thank you very much for your positive comments on our manuscript.

We did not intend to claim that "the mixed layer depth computed from the monthly averaged T and S is a totally adequate reflection of the monthly-mean of the MLD calculated from observational profiles". Rather, we have found that it was the best possible choice for our model intercomparison, because the MLDs calculated online in the OMIP models are impossible to compare with each other due to the different methods used by the modelling groups (as shown in table 1). We acknowledge that our wording probably failed to convey this meaning. We will clarify this in the revised version of the manuscript.

Your suggestion to use other models for Figure 6 is of course relevant, but we used the two models for which the datasets were accessible. Daily 3D fields of temperature and salinity have been published on ESGF for the IAP-LICOM high resolution model, although the retrieval of these fields is a challenge (which is the reason why only one year, 1998, was used). Daily 3D fields are available for FSU-HYCOM, but not published on ESGF. For Fig.6, Eric Chassignet and Alexandra Bozec have recomputed the MLD from one year of daily output, using the method we have chosen in our paper, in order to enable the comparison.

We agree that the use of a larger density jump when working with monthly model output could make differences such as shown in Fig.6 smaller for any given month, but it is unlikely that a single value of the density jump would effectively reduce the difference at all latitudes and for all months. As an example, the difference in MLD computed with the two density thresholds (0.03 and 0.125) is shown below for the ISAS climatology. Its seasonal cycle differs from the seasonal cycle of the daily-monthly MLD differences shown in our Fig.6. For this reason we do not wish to advocate this approach in our manuscript.

[Figure]

We will make the  minor corrections you suggest in the revised version of our manuscript.

---

## Author Response (AR1)

Review of "The Mixed Layer Depth in the Ocean Model Intercomparison Project (OMIP): Impact of Resolving Mesoscale Eddies" by Treguier et al.

**Reply to reviewer #1 (Steve Griffies)**

*This is a well-written summary of the comparison of mixed layers in a suite of OMIP2 simulations, including both coarse (1degree) and fine grids. The results are provocative and provide a benchmark and motivation for future work. I support publication and only have a few comments.*

Many thanks for your review of our manuscript and your encouraging comments.

*-line 26: We never really "validate" climate models. Instead we "evaluate" models.*
We have replaced "validation" by "evaluation".

*-line 74: "period period"*
Typo corrected.

*-line 188: "of course" is subjective; suggest removing*
We agree, "of course" has been removed.

*-In many many places in the manuscript, the word "resolution" is used when you really mean "grid spacing". Resolution is a non-dimensional number whereas grid spacing measures the distance between grid points, typically in degrees or km. In most places this quibble is not so important since we "know what is meant" even if we do not say it clearly. But on line 425 one reads the rather confusing sort of statement that can result when "resolution" and "grid spacing" are interchanged: "with a horizontal resolution of less than 1km". Does that phrase mean the grid spacing is coarser than 1km or finer than 1km?? This is the sort of confusion that a novice (and experienced) reader can come across when "resolution" is used when "grid spacing" is meant. I suggest clarifying all uses of "resolution".*

Thanks for this suggestion, we have modified the manuscript accordingly (line numbers refer to the first version of the manuscript):
- abstract, line 24, "1° grid spacing".
- line 95, replace "1°" by "1° grid spacing".
- line 99, replace "up to 1/16°" by "up to 1/16° grid spacing".
- line 142, replace "horizontal resolution" by "horizontal grid spacing"
- line 147, replace "0.1° resolution" by "0.1° configuration"
- line 148, replace "1° resolution" by "1° configuration"
- line 149 and line 150, replace "Both resolutions" by "Both configurations"
- line 151, replace "at 0.1° resolution" by "at 0.1° grid spacing"
- line 152, replace "at 1° resolution" by "at 1° grid spacing"
- line 158, replace "nominal resolution" by "nominal grid spacing"
- line 166, replace "1/16° horizontal resolution" by "1/16° horizontal grid spacing"
- line 180, replace "horizontal resolution" by "horizontal grid spacing".
- line 377, replace "a low resolution (1°) reanalysis" by "a reanalysis with coarse (1°) grid spacing"
- line 391, remove "high resolution" before "1/16° CMCC model"
- line 394, replace "to reach a 1° resolution" by "to a nominal 1° grid"
- line 425, replace "horizontal resolution of less than 1km" to "grid spacing of less than 1 km".

*-line 575: The authors observe that the MLD is more widely spread among the OMIP simulations than SST. This is a very important statement that perhaps has been noted before but is worth emphasizing more. It offers an important counter-point to those who discount OMIP simulations for using a prescribed atmosphere and so "have the answer given to them". There is a lot more physics going into the MLD than just that provided by the SST and SSS.*

Thanks for this interesting comment. We have added two sentences to emphasize our statement more, line 578 in section 6:
**" It is important to note that the mixed layer heat content, an important variable for climate, is not well constrained by the SST alone and that the mixed layer depth depends on internal ocean dynamics. As noted by Griffies et al., (2009) and Danabasoglu et al. (2014), OMIP experiments reveal the key influence of these ocean dynamics (and their representation in models) on major climate-relevant processes such as mixed layer properties, water mass subduction and meridional overturning**."

*-Now for my slightly nontrivial and somewhat self-serving comment. Namely, the authors point to the sensitivity of the MLD to the density threshold (Figure 2), the upper layer depth, and time sampling. In the end, they propose a 10m upper layer starting point rather than the "top grid cell" advised by Griffies et al (2016). I think this is a good suggestion. Yet they also support the density threshold approach, even though it suffers from the problems they note in their Figure 2, as well as those problems noted by BM04 whereby the density threshold should be a function of the SST/SSS given the nonlinear equation of state. These limitations and hyper-sensititivities motivated Reichl et al (2022) to propose a potential energy-based threshold. That approach also can use monthly mean T/S to directly compare model to obs, and it can be implemented online. So it is a practical approach and simpler than some methods like Holte and Talley, though more complex than the density threshold method.*

*I do not ask the authors to add the energy approach to their analysis, as that would be more work than I can reasonably request. But I do suggest they be somewhat more circumspect about the density threshold for future MIPs. I might be wrong, but at this point I think a potential energy approach ala Reichl et al is a physically compelling and practical approach that avoids many of the problems with the density approach.*

We agree that we could mention in more detail the new approach presented by Reichl et al. As you point out, more work is necessary to fully implement the potential energy method at the global scale, and this is beyond the scope of our paper. We have added the following sentence line 219:
**"The potential energy-based method proposed by Reichl et al. (2022) may have the advantage of being less sensitive to the model's vertical resolution than other methods, but more research is needed to understand how to choose a potential energy threshold, and whether it is possible to define one that would be relevant globally and for all seasons**".

**Reply to reviewer #2 (George Nurser)**

*Major comments*

*This is an important and timely piece of work on a key topic that is only going to become more important for climate science. The work presented here is rigorous and careful, and the authors have taken care to point out the inconsistencies in the model datasets employed here —there are not many eddy resolving runs available forced by the same atmospheric forcing datasets, and different vertical resolutions and parameterisations are used by different models. Hopefully this MS will stimulate further work on this subject.*

*The suggested definition of the model MLD as the shallowest depth where as $\rho - \rho 10m \geq 0.03$ kg m-3 (at least when calculating MLD online) should be followed in future work.*

*It is an important finding that finding that (generally) the MLD thus defined is indeed shallower at eddy-resolving resolution, presumably because the eddy parameterisations at coarse resolutions are not as effective at fluxing buoyancy upwards as are the resolved eddies at coarse resolution.*

*It is also interesting to see how sensitive the MLD are to the actual ML model and to model vertical resolution, especially in summer.*

We thank you very much for these positive comments on our manuscript.

*I was not fully convinced, however, of the argument that the MLD calculated from the monthly-average temperature and salinity, MLD(⟨T⟩, ⟨S⟩), was a totally adequate reflection of the monthly-mean of the MLD calculated online at each time step ⟨MLD(T, S)⟩, and thus a fully comparable field to the MLD calculated from individual observational profiles. For the zonally averaged ⟨MLD(T,S)⟩ − MLD(⟨T⟩, ⟨S⟩) in Fig. 6 seem to reach ~30–40 m in late spring. Moreover of the two models plotted in Fig. 6, IAP-LICOM is very much an outlier, and FSU-HYCOM has notably coarse vertical resolution (~ 36 layers) and so would not expect to show much of an effect, especially in summer.*

We did not want to claim that "the mixed layer depth computed from the monthly averaged T and S is a totally adequate reflection of the monthly-mean of the MLD calculated from observational profiles". Rather, we have found that it was the best possible choice for our model intercomparison, because the MLDs calculated online in the different models are impossible to compare with each other, due to the different methods (as shown in table 1). We have added a sentence to better explain our strategy,  line 387:
**"We acknowledge that computing MLDs from monthly averaged T and S is not satisfying, and that the only motivation to do so is to allow the intercomparison of models for which the MLDs computed online are not comparable with each other".**

*Fig. 6 would be more convincing if it included output from a higher resolution, "good" model, like CMCC-NEMO which has 98 levels. Why not use NCAR-POP and CMCC-NEMO as were used in Fig. 3 to show the impact of using different reference levels?*

Your suggestion to use other models for Figure 6 is of course relevant, but we used the two models for which the datasets were accessible. Daily 3D fields of temperature and salinity have been published on ESGF for the IAP-LICOM model, although the retrieval of these

fields is a challenge (which is the reason why only one year, 1998, was used). Daily 3D fields are available for FSU-HYCOM, but not published. For Fig.6, E. Chassignet and A. Bozec have recomputed the MLD from one year of daily output, using the method we have chosen in our paper, to enable the comparison.

*I agree that for purposes of comparability it was necessary to use MLD($\langle T \rangle$, $\langle S \rangle$), but suggest that it would have been better to have tried a larger density difference criterion when using monthly-mean averages. For it is conventional when calculating MLD using monthly-mean observational climatologies to use a larger density difference, e.g. WOA 2018, which uses $\rho - \rho 10m \geq 0.125$ kg m-3 s. It would be relatively straightforward to calculate MLD using monthly-mean T and S, with a range of density differences starting up from 0.03 kg m-3 and ranging up to 0.125 kg m-3 using e.g. CMCC-NEMO, and choose the density difference that minimised deviations from the monthly-average of the MLD calculated online with the 0.03 kg m-3 density criterion (i.e minimise the difference field plotted out in the panels in Fig. 6).*

We agree that the use of a larger density jump when working with monthly model output could make differences such as shown in Fig.6 smaller for any given month, but it is unlikely that a single value of the density jump would effectively reduce the difference at all latitudes and for all months. As an example, the difference in MLD computed with the two density thresholds (0.03 and 0.125) is shown below for the ISAS climatology. Its seasonal cycle differs from the seasonal cycle of the daily-monthly MLD differences shown in our Fig.6. For this reason we do not wish to advocate this approach in our manuscript.

[Figure]

*Recommendation*

*This manuscript will be very useful for the field and should be published subject to minor corrections.*

We thank the reviewer for this positive recommendation.

*Detailed comments*

*P6, l138. Does IAP-LICOM include the wave generated mixing of Qiao, et al., GRL, 2004?*
No, it does not.

*P6, l164 "NEMO version 3.64" Is this what you mean?*
Thanks for catching this error, we meant NEMO version 3.6. We have corrected the number.

*P14, Fig. 4. Caption. Observation datasets ⇒ observational datasets.*
Corrected.

*P17, Fig. 4. Why not use NCAR-POP and CMCC-NEMO for consistency?*

I suppose you mean Fig 6. Unfortunately, the daily 3D fields have not been saved for NCAR-POP, and we did not have the resources necessary for the re-computation of the MLD from the 3D daily fields stored at CMCC (see our answer to your comment on Fig 6 above).

---

## Author Response (AR2)

Revised version of "The Mixed Layer Depth in the Ocean Model Intercomparison Project (OMIP): Impact of Resolving Mesoscale Eddies"

**Message from editor:**

*Dear Authors,*
*as you can see both Reviewers recommend publication of your manuscript.*
*Checking the data availability section, I have noticed that the link for the data and notebooks seems to be broken:*
*"The data used to produce the figures and the corresponding python notebook are archived on Zenodo, doi:10.5281/zenodo.7656425"*
*Could you please double-check this DOI?*
*Thanks,*
*Riccardo*

**Reply:**

Dear Riccardo,
My apologies for the mistake. The link is https://doi.org/10.5281/zenodo.7656425. It has been corrected in the manuscript.
Best regards,
Anne Marie